# Xanthomonas immunity proteins protect against the *cis*-toxic effects of their cognate T4SS effectors

Gabriel U Oka [ID][1,2], Diorge P Souza[1,3], Germán G Sgro [ID][1,4], Cristiane R Guzzo[5], German Dunger [ID][1,6] & Chuck S Farah [ID][1✉]

## Abstract

**Many bacteria kill rival species by translocating toxic effectors into target cells. Effectors are often encoded along with cognate immunity proteins that could (i) protect against "friendly-fire" (*trans*-intoxication) from neighboring sister cells and/or (ii) protect against internal *cis*-intoxication (suicide). Here, we distinguish between these two mechanisms in the case of the bactericidal *Xanthomonas citri* Type IV Secretion System (X-T4SS). We use a set of *X. citri* mutants lacking multiple effector/immunity protein (X-Tfe/X-Tfi) pairs to show that X-Tfis are not absolutely required to protect against *trans*-intoxication by wild-type cells. Our investigation then focused on the in vivo function of the lysozyme-like effector X-Tfe[XAC2609] and its cognate immunity protein X-Tfi[XAC2610]. In the absence of X-Tfi[XAC2610], we observe X-Tfe[XAC2609]-dependent and X-T4SS-independent accumulation of damage in the *X. citri* cell envelope, cell death, and inhibition of biofilm formation. While immunity proteins in other systems have been shown to protect against attacks by sister cells (*trans*-intoxication), this is an example of an antibacterial secretion system in which the immunity proteins are dedicated to protecting cells against *cis*-intoxication.**

**Keywords** Type IV Secretion System; *cis*-intoxication; Bacterial Competition; Biofilm; Immunity Protein
**Subject Categories** Immunology; Microbiology, Virology & Host Pathogen Interaction; Signal Transduction

## Introduction

Bacteria are continuously competing with each other for space and nutrients. One widespread competition strategy is the secretion of effectors into adjacent bacterial cells (Russell et al, 2011; Benz and Meinhart, 2014; Souza et al, 2015; Cianfanelli et al, 2016) mediated by specialized secretion systems, as for example: the Type IV Secretion Systems (T4SSs) from *Xanthomonas citri* (Souza et al, 2015) and *Stenotrophomonas maltophilia* (Bayer-Santos et al, 2019), the Type V Secretion System (T5SS) from *Escherichia coli* (Aoki et al, 2005), the Type VI Secretion Systems (T6SSs) from *Pseudomonas aeruginosa* (Russell et al, 2011), *Burkholderia thailandensis* (Schwarz et al, 2010), *Salmonella typhimurium* (Sana et al, 2016), *Vibrio cholerae* (MacIntyre et al, 2010) and *Serratia marcescens* (Murdoch et al, 2011), the *Caulobacter crescentus* CDZ-based Type I secretion System (García-Bayona et al, 2017), and the Type VII Secretion Systems (T7SSs) encoded by Gram-positive bacteria *Staphylococcus aureus* (Cao et al, 2016) and *Bacillus subtilis* (Tassinari et al, 2022; Kobayashi, 2021). All these different secretion systems have at least one effector and one cognate immunity protein, the latter of which protects the donor from self-intoxication.

T4SSs are large protein complexes that traverse the cell envelope, producing a channel through which proteins or protein–DNA complexes can be secreted into animal or plant hosts or other bacterial cells (Alvarez-Martinez and Christie, 2009; Ilangovan et al, 2015; Li and Christie, 2018; Sheedlo et al, 2022). Canonical type-A T4SSs are usually composed of 12 proteins, VirB1-VirB11 and VirD4 (Fronzes et al, 2009; Alvarez-Martinez and Christie, 2009; Costa et al, 2015). Chromosome-encoded T4SSs found in the order Xanthomonadales and some other proteobacterial species (X-T4SSs) secrete antibacterial effectors (X-Tfes) that are recruited via interactions with the VirD4 ATPase coupling protein (Alegria et al, 2005; Souza et al, 2011; Oka et al, 2022; Sgro et al, 2019). All X-Tfes contain a conserved C-terminal domain termed XVIPCD that interacts directly with the all-alpha domain of VirD4 (Alegria et al, 2005; Oka et al, 2022). In the phytopathogen *Xanthomonas citri*, the X-Tfe[XAC2609] effector is secreted in a manner that is dependent on its XVIPCD and on a functional X-T4SS (Souza et al, 2015; Oka et al, 2022). The N-terminal portion of X-Tfe[XAC2609] contains a glycoside hydrolase family 19 (GH19) domain that cleaves peptidoglycan (PG) and a PG binding domain. This PG hydrolase activity is inhibited by its cognate immunity protein X-Tfi[XAC2610] (Souza et al, 2015). Therefore, X-Tfe[XAC2609] and X-Tfi[XAC2610] form an effector/immunity protein pair associated with the *X. citri* X-T4SS, which provides an adaptive advantage for X. citri in co-cultures with *E. coli* and other bacterial species (Oka et al, 2022; Souza et al, 2015).

[1]Departamento de Bioquímica, Instituto de Química, Universidade de São Paulo, São Paulo, SP, Brazil. [2]Structure and Function of Bacterial Nanomachines, Institut Européen de Chimie et Biologie—CNRS, UMR 5234 Microbiologie Fondamentale et Pathogénicité University of Bordeaux, Pessac, France. [3]Division of Cell Biology, MRC Laboratory of Molecular Biology, Cambridge, UK. [4]Departamento de Ciências BioMoleculares, Faculdade de Ciências Farmacêuticas de Ribeirão Preto, Universidade de São Paulo, Ribeirão Preto, SP, Brazil. [5]Departamento de Microbiologia, Instituto de Ciências Biomédicas, Universidade de São Paulo, São Paulo, SP, Brazil. [6]Instituto de Ciencias Agropecuarias del Litoral (ICiAgro Litoral), Universidad Nacional del Litoral, CONICET, Facultad de Ciencias Agrarias, Esperanza, Argentina. ✉E-mail: chsfarah@iq.usp.br

Secretion system-mediated bacterial killing is typically evaluated in interbacterial competition assays between prey and attacker cells that code for one or more different effector-immunity protein pairs. The rationale is that the prey is susceptible to the toxicity of the delivered effector(s) because they do not produce at least one cognate immunity protein (Aoki et al, 2005; Russell et al, 2011, 2012; García-Bayona et al, 2017; Kobayashi, 2021; Tassinari et al, 2022). Since most bacteria–bacteria interactions are between genetically identical cells (e.g., bacterial colonies), bacteria coding for toxic effectors also code for immunity proteins that protect themselves against intoxication via their own effectors. It is reasonable to suppose that immunity proteins should be localized in the same subcellular compartment where its cognate effector acts (Benz and Meinhart, 2014; Whitney et al, 2013; Russell et al, 2014; Jurėnas and Journet, 2021); for example, the X-Tfe[XAC2609] toxin that targets peptidoglycan has a cognate immunity protein, X-Tfi[XAC2610], that carries an N-terminal signal peptide and lipobox that directs it to the periplasm (Souza et al, 2015; Sgro et al, 2019). In this scenario, immunity proteins could, in principle, provide protection against two different, not necessarily exclusive, toxicity mechanisms: (i) intoxication due to "friendly-fire" translocation of toxic effectors from neighboring sister cells (fratricide or *trans*-intoxication; Fig. 1A) or (ii) self-intoxication (suicide or *cis*-intoxication; Fig. 1B) that results from the action of endogenously produced toxins. Both *cis*- and *trans*-intoxication mechanisms have been observed previously for T6SS-mediated effector transfer (Basler and Mekalanos, 2012; Hood et al, 2010; Russell et al, 2011; Dong et al, 2013; Li et al, 2012; Whitney et al, 2015).

Here, we show that an *X. citri* strain lacking multiple effector-immunity protein pairs remains resistant to fratricidal X-T4SS-mediated attack by wild-type *X. citri* cells, thus providing evidence that the role of X-T4SS immunity proteins (X-Tfis) is not restricted to avoiding X-T4SS-mediated fratricide (*trans*-intoxication). We also show that an *X. citri* X-Tfi[XAC2610] knockout strain suffers autolysis in a process that is mediated by X-Tfe[XAC2609] but is independent of the X-T4SS. Cell ultra-structural aspects of the autolysis process were analyzed by fluorescence and electron microscopies. We demonstrate that X-Tfi[XAC2610] is important for biofilm formation and is required to protect against the detrimental effects of X-Tfe[XAC2609], even in the absence of a functional X-T4SS. These results support the conclusion that the protective function of X-Tfis is geared mainly towards the *cis*-intoxication (self-intoxication) effects of the endogenous X-Tfes.

# Results

## The X-T4SS immunity proteins are not the primary defense against *trans*-intoxication (fratricide)

*X. citri* is able to kill *E. coli* in an X-T4SS-dependent manner (Fig. 2A; Appendix Fig. S1), as has been previously shown (Souza et al, 2015; Oliveira et al, 2016; Oka et al, 2022; Sgro et al, 2018). *Δ8Δ2609-GFP* is an *X. citri* strain in which the genes for eight X-Tfe/X-Tfi pairs (XAC2885/XAC2884, XAC0574/XAC0573, XAC0096/XAC0097, XAC3634/XAC3633, XAC1918/XAC1917, XAC0466/XAC0467, XAC4264/XAC4263/XAC4262, XAC3266/XAC3267) were deleted and a ninth X-Tfe gene (that codes for

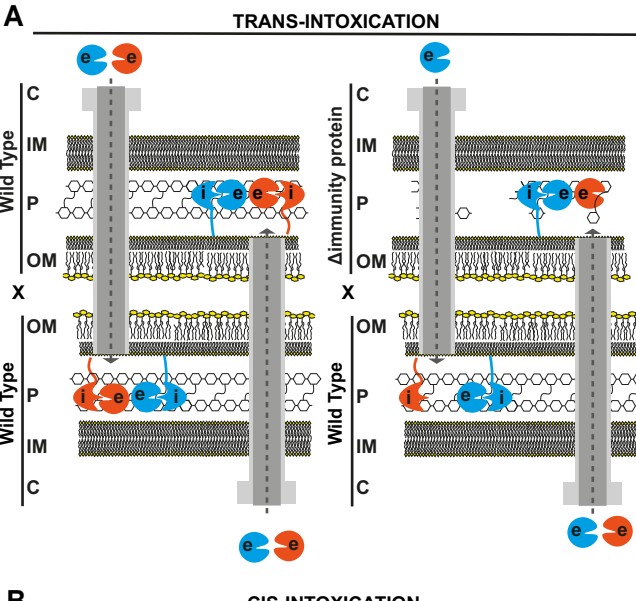

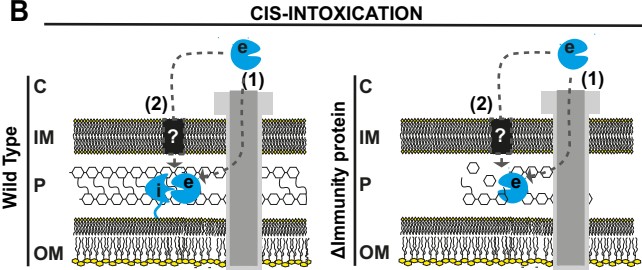

**Figure 1. Schematic model of *trans*- and *cis*-intoxication mechanisms.**

(A) *Trans*-intoxication. In this mechanism, intoxication is due to contact- and T4SS-dependent transfer of X-Tfes (effectors) from one cell to another. Left: Genetically identical cells with equivalent repertoires of X-Tfes and cognate X-Tfis (immunity proteins) would be protected. In the scheme shown here, two wild-type cells that produce two different effector-cognate immunity protein pairs (orange and blue; e: effectors and i: immunity proteins) are immune against the toxic effects of the X-T4SS-mediated *trans*-intoxication due to the protective role of cognate immunity proteins. Right: Encounters between cells with non-equivalent repertoires would lead to killing. In the scheme shown here, a wild-type cell that produces two different effector-cognate immunity protein pairs (blue and orange) is in contact with a mutant cell that produces only one effector-immunity protein pair (blue). The hypothesis is that the two cells transfer effectors into each other's periplasm and since the prey cell lacks the immunity protein that inhibits the orange effector, its cell wall is susceptible to degradation. (B) *Cis*-intoxication. In this mechanism, instead of being transported outside of the cell by the X-T4SS, an effector is translocated into the periplasm. Translocation could be T4SS-dependent (1) or T4SS-independent (2). Left: A wild-type cell carrying a complete set of cognate immunity proteins is protected against self-intoxication. Right: A bacterial strain lacking the immunity protein (ΔImmunity protein), may be susceptible to the cumulative activity of an effector that leaks into the periplasm. C cytoplasm, IM inner membrane, OM outer membrane, P periplasm.

X-Tfe[XAC2609]) was substituted with the coding sequence for green fluorescent protein (GFP) (Bayer-Santos et al, 2019; Oka et al, 2022). Figure 2A shows CPRG-based colorimetric assays to quantitatively monitor real-time interspecies killing of *E. coli* cells in *X. citri/E. coli* co-cultures. As previously shown, *Δ8Δ2609-GFP* and the X-T4SS-defective *ΔvirB7* strain do not kill *E. coli* (Bayer-Santos et al, 2019; Oka et al, 2022), demonstrating that the X-T4SS

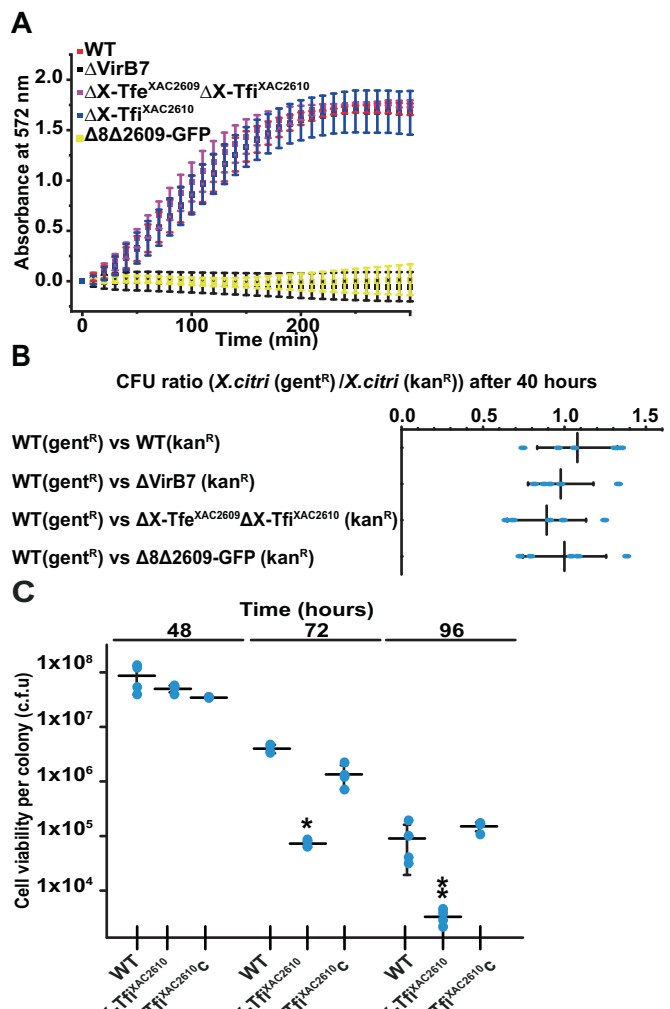

**A**

**B**

CFU ratio (*X.citri* (gent$^R$)/*X.citri* (kan$^R$)) after 40 hours

**C**

**Figure 2.** **X-T4SS-related immunity proteins are not required to confer protection against *trans*-intoxication mediated by the X-T4SS.**

(A) Bacterial competition of *X. citri* against *E. coli* MG1655 cells expressing β-galactosidase. *E. coli* killing was monitored by detecting the degradation of chlorophenol red-β-D-galactopyranoside (CPRG) by measuring absorbance at 572 nm at 10 min intervals. Data points present the means +/− SD of three experiments (biological replicates). (B) *X. citri* cell viability ratio (*X.citri*(gent$^R$)/ *X.citri*(kan$^R$)) after co-culture in Luria-Bertani (LB) agar after 40 h. *X. citri* strains used: wild-type (WT), Δ*virB7* (ΔVirB7), ΔX-Tfe$^{XAC2609}$ΔX-Tfi$^{XAC2610}$ and *Δ8Δ2609-GFP*. In each co-culture experiment, the WT strain carried the pBBR(MCS5)GFP plasmid conferring resistance to gentamicin (gent$^R$) and the genetically modified strain carried the pBBR(MCS2)RFP vector that confers resistance to kanamycin (kan$^R$). Colony-forming units per mL (CFU/mL) of each strain was assessed through serial dilution assays on LB-agar plates carrying the appropriate antibiotics. Mean (vertical bars) +/− SD (horizontal bars); $n = 5$ (biological replicates) with each replicate represented by a single data point (blue dot). Analysis of variance (Anova) *P* value = 0.66, thus at 0.05 level, the values are not significantly different. (C) Colony viability assay. *X. citri* wild-type strain (WT, ΔX-Tfi$^{XAC2610}$ strain and ΔX-Tfi$^{XAC2610}$ + X-Tfi$^{XAC2610}$ strain (ΔX-Tfi$^{XAC2610}$C) were grown on LB-agar plates. After 48 h, 72 h, and 96 h at 30 °C, the colonies were resuspended in 2xTY media, and cellular viability (CFU/mL) was assessed through serial dilution assays on LB-agar plates. Mean (horizontal bars) +/− SD (vertical bars); $n = 4$ (biological replicates). Each replicate is represented by a single data point (blue dot). *Unpaired *t* test *P* values of the data at 72 h are <0.0001 for WT vs ΔX-Tfi$^{XAC2610}$ and <0.01 for ΔX-Tfi$^{XAC2610}$C vs ΔX-Tfi$^{XAC2610}$. **Unpaired *t* test *P* values of the data at 96 h are 0.056 for WT vs ΔX-Tfi$^{XAC2610}$ and <0.0001 for ΔX-Tfi$^{XAC2610}$C vs ΔX-Tfi$^{XAC2610}$. Source data are available online for this figure.

to neutralize exogenous effectors that were injected by neighboring bacteria (*trans*-intoxication scheme, Fig. 1A).

## X-Tfi$^{XAC2610}$ provides immunity against in vivo intracellular autolytic activity of X-Tfe$^{XAC2609}$

As the above results are inconsistent with the proposition that X-Tfis provide protection against *trans*-intoxication, we performed experiments to test the hypothesis that they instead provide protection against self-intoxication (*cis*-intoxication), as illustrated in Fig. 1B. In the case of effectors that normally act in the periplasm of target cells (for example, lysozyme-like effectors), *cis*-intoxication may arise if the effector is transferred across the inner membrane and into the periplasm of the producing cell by one or more routes, either dependent or independent of the X-T4SS. To test this hypothesis, we used *X. citri* strains with single or multiple in-frame deletions of genes coding for the X-Tfe$^{XAC2609}$ lysozyme-like effector, its cognate immunity protein X-Tfi$^{XAC2610}$ and the VirB7 and VirD4 subunits that are essential for X-T4SS function.

Appendix Fig. S2 shows that colonies of *X. citri* wild-type, ΔX-Tfi$^{XAC2610}$, ΔX-Tfi$^{XAC2610}$Δ*virB7*, ΔX-Tfi$^{XAC2610}$c (c: complementation with an extrachromosomal plasmid expressing X-Tfi$^{XAC2610}$), and ΔX-Tfi$^{XAC2610}$cΔ*virB7* grown on LB-agar plates for 24 and 48 h are indistinguishable in terms of color, roughness, opacity and size. However, after 72 h of growth, colonies of all strains with X-Tfe$^{XAC2609}$ but lacking X-Tfi$^{XAC2610}$ (ΔX-Tfi$^{XAC2610}$, ΔX-Tfi$^{XAC2610}$Δ*virB7*) became partially transparent, indicative of cell death. On the other hand, the lineages lacking an X-Tfe$^{XAC2609}$ gene or carrying an X-Tfi$^{XAC2610}$ gene all maintained their opacity at 72 h (including strains ΔX-Tfi$^{XAC2610}$c, ΔX-Tfi$^{XAC2610}$cΔ*virB7* that carry a plasmid that confers expression of full-length X-Tfi$^{XAC2610}$). Importantly, no reduction in colony opacity was observed after

presents antibacterial activity and that this activity is dependent on the presence of a cohort of secreted effectors (X-Tfes). On the other hand, the deletion of X-Tfi$^{XAC2610}$ or the single X-Tfe$^{XAC2609}$/X-Tfi$^{XAC2610}$ pair does not significantly impair the antibacterial function of the X-T4SS under the conditions tested. In order to test whether the *trans*-intoxication (fratricide) hypothesis (Fig. 1A) is valid for the X-T4SS, we performed intraspecies bacterial competition assays using wild-type *X. citri* against its derivative mutants. Figure 2B shows that the functional X-T4SS in the wild-type strain fails to confer a competitive advantage against the *X. citri* Δ*virB7* strain that lacks a functional X-T4SS due to the absence of the VirB7 subunit (Souza et al, 2015; Oliveira et al, 2016; Sgro et al, 2018). This result itself is not inconsistent with the *trans*-intoxication hypothesis since the target *X. citri* Δ*virB7* strain still carries a full set of X-Tfis. However, wild-type *X. citri* cells were also unable to kill the ΔX-Tfe$^{XAC2609}$/ΔX-Tfi$^{XAC2610}$ double-mutant *X. citri* strain which lacks the X-Tfe$^{XAC2609}$/X-Tfi$^{XAC2610}$ toxin-antitoxin pair (Fig. 2B) nor do they kill the *Δ8Δ2609-GFP X. citri* strain in which eight other X-Tfe/X-Tfi effector/immunity protein pairs were deleted. The observation that wild-type *X. citri* is unable to kill the *X. citri* ΔX-Tfe$^{XAC2609}$ΔX-Tfi$^{XAC2610}$ or *Δ8Δ2609-GFP* strains indicates that the primary role of X-T4SS immunity proteins is not

72 h of growth for the double mutant ΔX-Tfe$^{XAC2609}$ΔX-Tfi$^{XAC2610}$, indicating that X-Tfe$^{XAC2609}$ must be present for the development of the observed phenotype (Appendix Fig. S2A). Interestingly, a cytosolic version of X-Tfi$^{XAC2610}$His-22-267 protects against X-Tfe$^{XAC2609}$ toxicity, suggesting that complex formation in the cytoplasm may impede leakage of the effector into the periplasm (Appendix Fig. S2C; ΔX-Tfi$^{XAC2610}$Cyt). Finally, transparent colonies are observed even in the absence of VirB7 (ΔvirB7), demonstrating that this phenotype does not depend on a functional X-T4SS, but rather results from the absence of the immunity gene. The colony opacity/transparency phenotypes were further validated using a convolutional neural network (CNN) analysis (Appendix Fig. S2).

All genes studied in this manuscript are located in a single genomic locus, and knockouts could potentially interfere with protein expression levels of nearby genes. To discard this possibility, western blot assays were performed. These experiments showed that only the expression levels of the targeted genes are affected by the genetic manipulations (Appendix Fig. S2B).

Figure 2C shows that after 48 h of growth on LB agar, no significant difference in cell counting or viability between wild-type and ΔX-Tfi$^{XAC2610}$ strains was observed. However, after 72 and 96 h of growth, the ΔX-Tfi$^{XAC2610}$ strain shows lower viability compared to the wild-type strain. The viability of the ΔX-Tfi$^{XAC2610}$ strain was restored when complemented with an extrachromosomal plasmid expressing X-Tfi$^{XAC2610}$.

In conclusion, these results indicate that (i) the transparent colony phenotype is due to the activity of X-Tfe$^{XAC2609}$, (ii) it can be inhibited by the presence of X-Tfi$^{XAC2610}$, and (iii) the phenotype does not require a functional X-T4SS. These data are consistent with a *cis*-intoxication mechanism by X-Tfe$^{XAC2609}$ that is inhibited by X-Tfi$^{XAC2610}$ as schematized in Fig. 1B.

## X-Tfi$^{XAC2610}$ is important for maintaining the integrity of the *X. citri* cell envelope

The results described above suggest that, in the absence of its cognate immunity protein, the endogenously produced X-Tfe$^{XAC2609}$ lysozyme-like effector induces cell autolysis. To test this hypothesis, we performed a set of time-lapse microscopy assays (Fig. 3; Movies EV1–5) using wild-type and *X. citri* mutant strains grown in media supplemented with propidium iodide (PI), a fluorescent dye that only stains nucleic acids when cell envelope integrity is compromised. We observed a significant increase in PI permeability in ΔX-Tfi$^{XAC2610}$ cells (Fig. 3; Movie EV2; Appendix Table S1) compared to wild-type cells (Fig. 3; Movie EV1; Appendix Table S1). Permeability is reduced to wild-type levels in the ΔX-Tfe$^{XAC2609}$ΔX-Tfi$^{XAC2610}$ double-mutant strain (Fig. 3; Movie EV3; Appendix Table S1). Transformation of ΔX-Tfe$^{XAC2609}$ΔX-Tfi$^{XAC2610}$ with a plasmid over-expressing the catalytic N-terminal domain of X-Tfe$^{XAC2609}$ (1–306), which lacks the XVIPCD X-T4SS secretion signal, greatly increased the amount of PI-permeable cells (Fig. 3; Movie EV4; Appendix Table S1). Moreover, multiple cell autolysis events were observed for the ΔX-Tfi$^{XAC2610}$ΔvirD4 *X. citri* strain (Fig. 3; Movie EV5; Appendix Table S1), confirming that toxicity caused by X-Tfe$^{XAC2609}$ does not require the XVIPCD secretion signal and is not mediated by the X-T4SS. Close inspection of Movies EV2, EV4, EV5, and EV6 and Fig. 4A shows that PI permeability of *X. citri* cells coincides with a rapid change in cell morphology, from natural rod to spherical, consistent with

X-Tfe$^{XAC2609}$-induced weakening of the cell wall. The onset of PI permeability was observed to occur both in isolated cells and in cells in contact with neighbors (Movie EV6; Fig. 4A). Another interesting observation is that the onset of PI permeability was frequently observed in cells that were undergoing cell division (Movie EV6), perhaps coinciding with a phase in the cell cycle where peptidoglycan integrity is more susceptible to the deleterious hydrolytic activity of X-Tfe$^{XAC2609}$.

We then used transmission electron microscopy (TEM) in order to obtain a more detailed picture of the differences in the cell envelope ultrastructure of *X. citri* wild-type and ΔX-Tfi$^{XAC2610}$ cells (Fig. 4B). TEM analyses of thin sections of previously fixed *X. citri* cells embedded in resin were used to assess structural details of the plasma membrane, cell wall, and intracellular content. TEM micrographs of *X citri* ΔX-Tfi$^{XAC2610}$ cells showed that they were frequently broken open with leakage of filamentous materials or were devoid of cellular contents (Fig. 4B; Appendix Figs. S3 and S4). In contrast, wild-type cells typically present an intact cell envelope and a high-density intracellular environment (Fig. 4B; Appendix Figs. S3 and S4). Micrographs of the ΔX-Tfi$^{XAC2610}$ΔvirB7 strain presented a phenotype similar to that observed for ΔX-Tfi$^{XAC2610}$ while micrographs of ΔvirB7 and ΔX-Tfi$^{XAC2610}$c lineages showed mostly intact cells as observed for the wild-type strain (Fig. 4B; Appendix Fig. S4). Analysis of the number of intact versus damaged cells in TEM micrographs indicates significant statistical differences in the percentage of lysed cells in strains producing both X-Tfe$^{XAC2609}$ and X-Tfi$^{XAC2610}$ (wild-type, ΔVirB7 and ΔX-Tfi$^{XAC2610}$c) compared with cells expressing the former but not the latter (ΔX-Tfi$^{XAC2610}$ and ΔX-Tfi$^{XAC2610}$ΔvirB7; Fig. 4C; Appendix Table S2). Together, fluorescence microscopy and TEM analyses indicate that, in the absence of X-Tfi$^{XAC2610}$, the activity of X-Tfe$^{XAC2609}$ promotes damage of the *X. citri* cell envelope, regardless of the presence of a functional X-T4SS.

## The hydrolytic activity of X-Tfe$^{XAC2609}$ inhibits *X. citri* biofilm formation in the absence of X-Tfi$^{XAC2610}$

The results so far show that the absence of X-Tfi$^{XAC2610}$ compromises the integrity of the *X. citri* cell envelope of only a small number of actively dividing cells during the exponential growth phase in liquid medium (Fig. 4C) and eventually leads to the lysis of a significant fraction of the cell population during stationary phase (Appendix Fig. S2). Since bacterial biofilms are a complex array of cells and extracellular polymers that develop over extended periods of time (many hours to days) with a progressive reduction in the rate of cell division (Dunger et al, 2014, 2016; Sena-Vélez et al, 2016), we decided to investigate the possible physiological effects associated with the deletion of X-Tfi$^{XAC2610}$ on biofilm formation. Figure 5A shows that, unlike the *X. citri* wild-type strain, ΔX-Tfi$^{XAC2610}$ and ΔX-Tfi$^{XAC2610}$ΔvirB7 cells could not form biofilms on polystyrene plastic surfaces after 5 days of cultivation in 2xTY medium. The double knockout strain ΔX-Tfe$^{XAC2609}$ΔX-Tfi$^{XAC2610}$ strain presents a normal biofilm while this double-mutant strain complemented with either full-length X-Tfe$^{XAC2609}$ or an N-terminal fragment lacking the XVIPCD secretion signal was defective in biofilm formation. On the other hand, the ΔX-Tfe$^{XAC2609}$ΔX-Tfi$^{XAC2610}$ strain complemented with the N-terminal domain of X-Tfe$^{XAC2609}$ carrying a detrimental mutation (E48A) in the active site of its GH19 PG hydrolase domain (Souza et al, 2015) was able to form a normal biofilm. These results

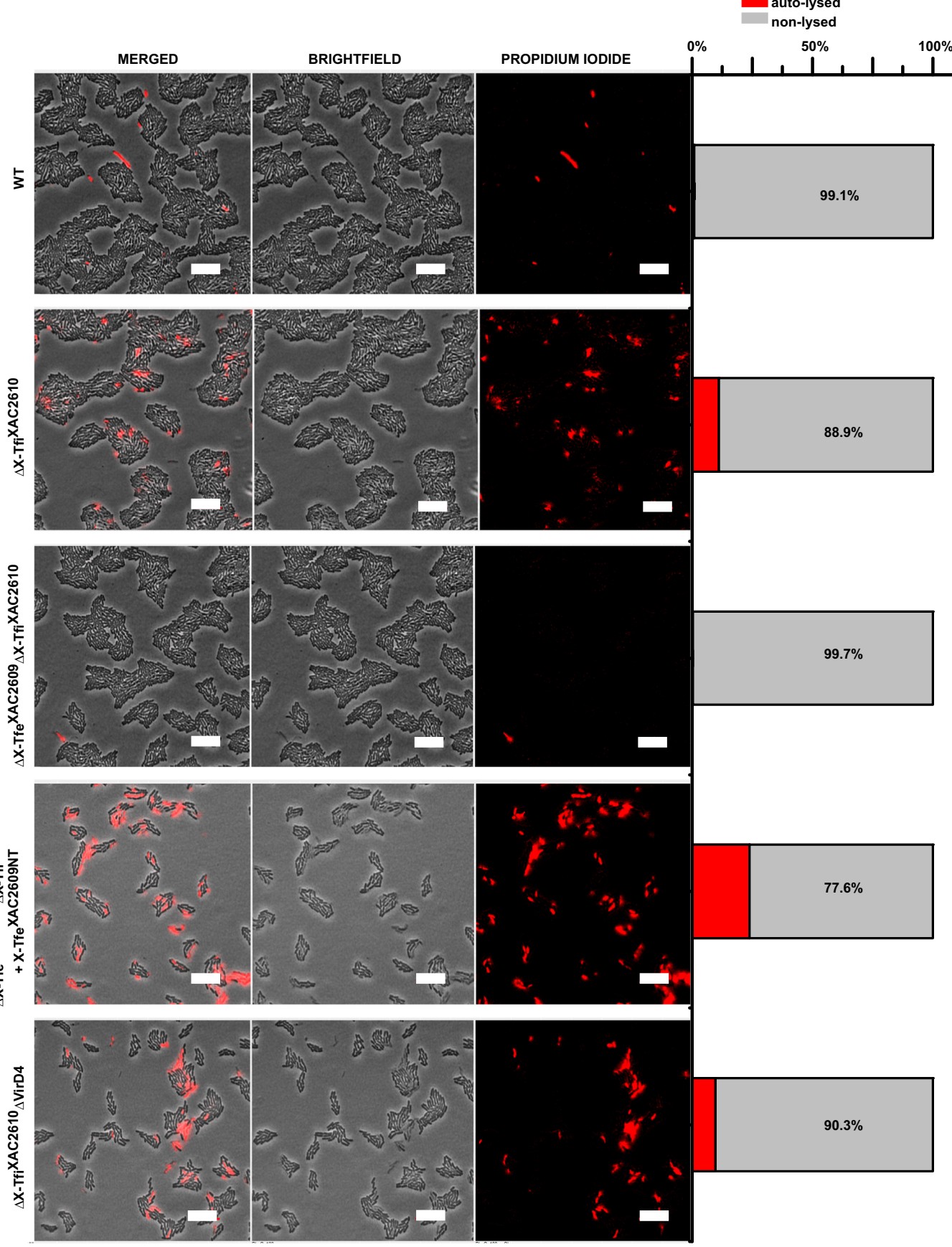

**Figure 3. Quantitative analysis of _X. citri_ cells exhibiting propidium iodide permeability shown in Movies EV1–5.**

Micrographs show the last time point of one of the two experiments shown in each Movie (parts A of Movies EV1–5). The first three columns show merged, bright-field and propidium iodide fluorescence images. Scale bar 10 µm. The final column presents the fraction % of cells that exhibit propidium iodide fluorescence (red, auto-lysed) and those that did not exhibit propidium iodide fluorescence (gray, non-lysed). Further details regarding this quantitative analysis are presented in Appendix Table S1. Source data are available online for this figure.

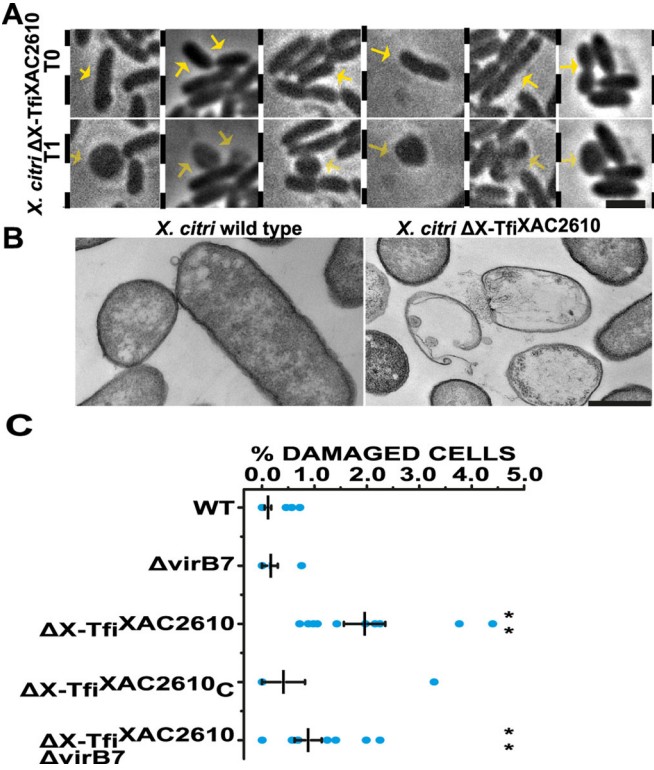

**Figure 4. X-Tfi^XAC2610 is important for maintaining _X. citri_ cell envelope integrity.**

(A) _X. citri_ ΔX-Tfi^XAC2610 strain time-lapse microscopy shows spontaneous spheroplasts forming events. Arrows at T0 point to cells that will turn into spheroplasts at T1. See Movie EV6 for more examples. Scale bar 2 µm. (B) Transmission electron microscopy (TEM) micrographs of _X. citri_ WT and ΔX-Tfi^XAC2610 strains after 12 h of growth in liquid 2xTY medium. Scale bars 500 nm. (C) Percentage of damaged cells observed in TEM micrographs. More examples of damaged cells are shown in Appendix Figs. S3 and S4. Total number of micrographs analyzed (_n_, technical replicates) for each _X. citri_ strain: wild-type (WT, _n_ = 15), ΔVirB7 (_n_ = 5), ΔX-Tfi^XAC2610 (_n_ = 10), ΔX-Tfi^XAC2610ΔVirB7 (_n_ = 10), ΔX-Tfi^XAC2610 complemented with X-Tfi^XAC2610 (ΔX-Tfi^XAC2610c, _n_ = 8). The percentage of damaged cells observed in each micrograph is represented by blue dots. Mean (vertical bars) +/− SEM (horizontal bars) are shown. ** denotes statistically significant differences in mean values when compared to the WT strain. Two sample _t_ test unpaired _P_ values for ΔX-Tfi^XAC2610 and ΔX-Tfi^XAC2610ΔVirB7 are $2.10 \times 10^{-5}$ and $3.41 \times 10^{-3}$, respectively. Further details regarding quantitative and statistical analysis are given in Appendix Table S2. Source data are available online for this figure.

confirm that, in the absence of its cognate inhibitory protein, the deleterious effects of X-Tfe^XAC2609 is dependent on its glycohydrolase activity.

We then asked if specific individual components of the X-T4SS are necessary for the translocation of X-Tfe^XAC2609 to the periplasm (Pathway (1) in Fig. 1B). This is a relevant hypothesis considering

that cytoplasmic substrates of the T4SS are known to interact with various T4SS subcomplexes along the secretion pathway (Cascales and Christie, 2004a; Atmakuri et al, 2004; Cascales and Christie, 2004b; Guzmán-Herrador et al, 2023). To address this question, we deleted the genes coding for several X-T4SS subunits that are associated with the bacterial inner membrane (VirB4, VirB6, VirB8, and VirD4) (Macé et al, 2022; Llosa et al, 2003), the outer membrane (VirB7, VirB9, and VirB10) (Fronzes et al, 2009; Chandran et al, 2009; Souza et al, 2011; Oliveira et al, 2016; Sgro et al, 2018) and the VirB5 subunit believed to form part of the extracellular pilus (Alvarez-Martinez and Christie, 2009; Christie et al, 2014; Sheedlo et al, 2022). These deletions were introduced in both the _X. citri_ wild-type and ΔX-Tfi^XAC2610 genetic backgrounds. Growth of these strains in 24-well plates for 24 h with agitation followed by 5 days without agitation revealed that all _X. citri_ strains lacking X-Tfi^XAC2610 cannot form biofilm, independent of the presence or absence of any X-T4SS structural component (Fig. 5B; Appendix Fig. S5). Taken together, these results show that, in the absence of X-Tfi^XAC2610, _cis_-intoxication by X-Tfe^XAC2609 inhibits biofilm formation in a manner that is independent of any subunit or subassembly of the X-T4SS apparatus.

Finally, as _X. citri_ is the causal agent of citrus canker, we asked whether the absence of X-Tfi^XAC2610 could affect the ability of this phytopathogen to cause disease. Appendix Fig. S6 shows that _X. citri_ ΔX-Tfi^XAC2610 can induce the appearance of citrus canker lesions in sweet orange leaves in a manner very similar to those caused by the wild-type strain. This phenotype was previously shown to be dependent on a type III secretion system (Cappelletti et al, 2011) but independent of the X-T4SS (Souza et al, 2011).

## Structural and co-evolutionary analyses of the X-Tfe^XAC2609/X-Tfi^XAC2610 complex

Figure EV1 shows the sequence conservation profile derived from the multiple sequence alignment of 429 non-redundant X-Tfi^XAC2610 homologs (Dataset EV1) and reveals several conserved motifs. One conserved motif is a region with several negatively charged amino acids corresponding to X-Tfi^XAC2610 residues 151–159 that form a $Ca^{2+}$-binding loop in the crystal structure of X-Tfi^XAC2610 (Souza et al, 2015). Appendix Fig. S7 shows that the presence of $Ca^{2+}$ ions significantly increases the thermal stability of X-Tfi^XAC2610. This result suggests that the family of X-Tfi^XAC2610 homologs could all be stabilized by divalent cation binding at this site.

To investigate the molecular mechanism of the immunity provided by X-Tfi^XAC2610 against X-Tfe^XAC2609 activity, we predicted the structure of the X-Tfe^XAC2609/X-Tfi^XAC2610 complex by Alpha-Fold2 (Mirdita et al, 2022; Varadi et al, 2022; Jumper et al, 2021) using the sequences of X-Tfi^XAC2610 lacking the N-terminal signal peptide (residues 54–267) and the N-terminal GH19 glycohydrolase domain of X-Tfe^XAC2609 (residues 1–194). Figure 6A shows that the best model of the X-Tfi^XAC2610(54–267)/X-Tfe^XAC2609(1–194)

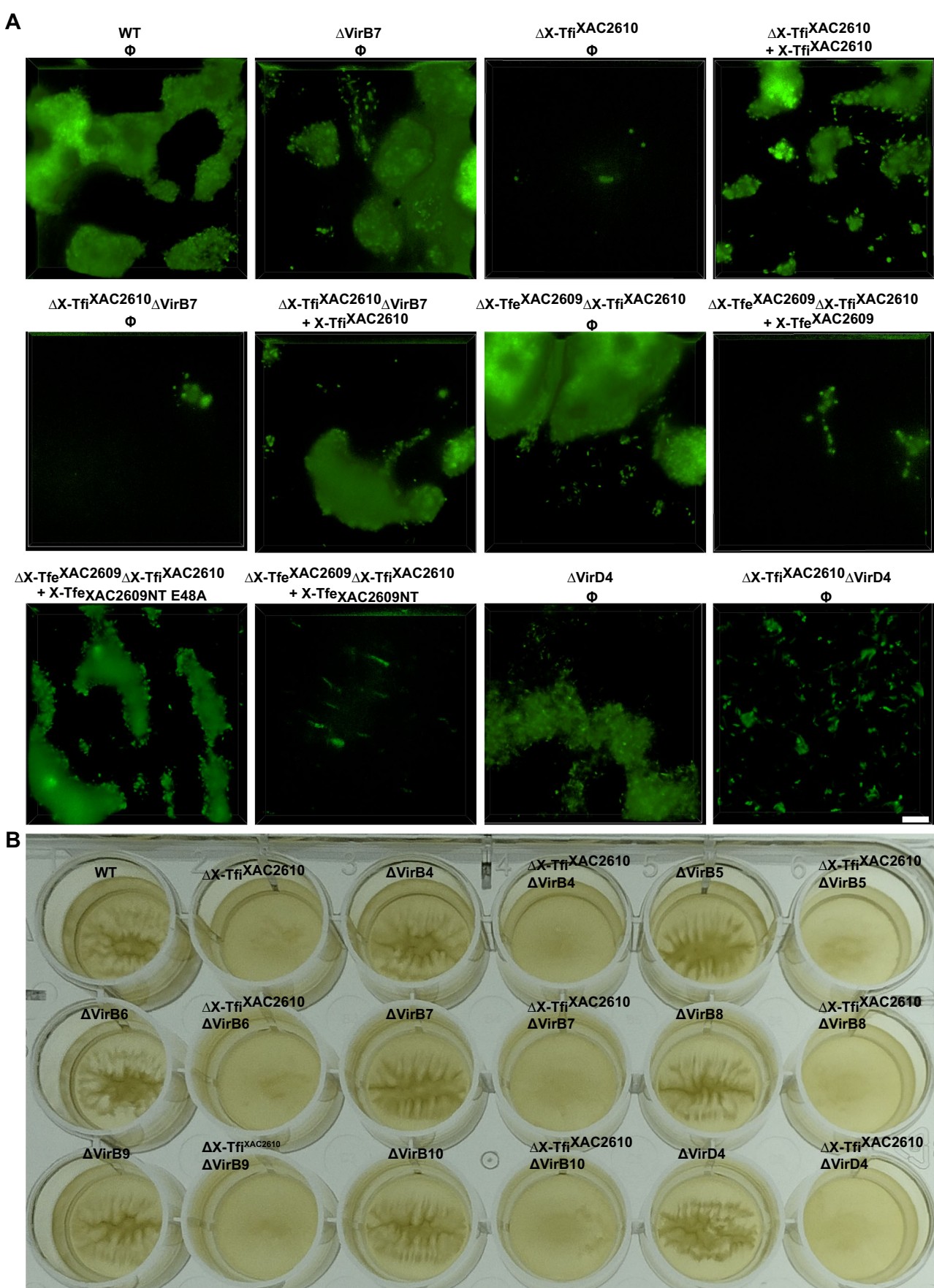

**Figure 5. The hydrolytic activity of X-Tfe^XAC2609 inhibits X. citri biofilm formation in the absence of X-Tfi^XAC2610.**

(A) *X. citri* wild-type and derived mutants carrying a plasmid for the endogenous expression of GFP were grown on 2xTY medium for 5 days at 30 °C in chambered microscope slides. *X. citri* wild-type (WT) strain, ΔVirB7 strain, ΔX-Tfi^XAC2610 strain, ΔX-Tfi^XAC2610ΔVirB7 strain, ΔX-Tfe^XAC2609ΔX-Tfi^XAC2610 strain containing the empty vectors pBRA (Φ) and complemented strains (ΔX-Tfi^XAC2610 + X-Tfi^XAC2610, ΔX-Tfi^XAC2610ΔVirB7 + X-Tfi^XAC2610, ΔX-Tfe^XAC2609ΔX-Tfi^XAC2610 + X-Tfe^XAC2609, ΔX-Tfe^XAC2609ΔX-Tfi^XAC2610 + X-Tfe^XAC2609NT, ΔX-Tfe^XAC2609ΔX-Tfi^XAC2610 ΔX-Tfe^XAC2609NT E48A) are indicated above each fluorescence microscopy image. Images were taken with a fluorescence microscope at ×100 magnification. Scale bars, 5 μm. (B) Biofilm formation assay in 24-well plates. Plates containing start cultures of *X. citri* were grown in 2xTY medium for 24 h at 30 °C at 200 rpm and then maintained at room temperature (22 °C) without shaking for 7 days. *X. citri* strains with knockouts in the genes for ΔX-Tfi^XA2610, one of the X-T4SS components structural components (VirB4 to VirB10 and VirD4) and the double knockouts were used and are indicated above each well. Source data are available online for this figure.

complex superposes with the previously determined crystal structure of X-Tfi^XAC2610 (Souza et al, 2015). In this model, the interface between the two proteins involves another well-conserved motif in the X-Tfi^XAC2610 family that includes a loop made up of residues 165–173 (Figs. 6A,B and EV1; Appendix Fig. S8). In the predicted X-Tfe^XAC2609/X-Tfi^XAC2610 complex, a highly conserved tyrosine (Y170) in this loop of X-Tfi^XAC2610 directly interacts with the catalytic aspartate (E48) in the active site of X-Tfe^XAC2609 (Figs. 6C and EV2; Appendix Fig. S8). To test the hypothesis that Y170 is involved in the mechanism of X-Tfi^XAC2610-mediated inhibition of X-Tfe^XAC2609, we carried out in vitro peptidoglycan hydrolysis assays using X-Tfe^XAC2609 in the absence or presence of wild-type X-Tfi^XAC2610 or the X-Tfi^XAC2610Y170A mutant. Figure 6D shows that the wild-type version of X-Tfi^XAC2610 inhibits the hydrolytic activity of X-Tfe^XAC2609, while the Y170A mutation significantly compromises X-Tfi^XAC2610 inhibition of peptidoglycan degradation by X-Tfe^XAC2609.

## Discussion

Peptidoglycan is a major component of the bacterial cell wall, functioning to maintain the stability of the cell envelope against turgor pressure (Callewaert et al, 2012; Silhavy et al, 2010; Navarro et al, 2022). By far, the most well-studied PG hydrolase is lysozyme, which was the first natural antimicrobial molecule isolated from the human body (Fleming and Wright, 1922). It targets the bacterial cell wall by hydrolyzing $\beta(1 \rightarrow 4)$ bonds between *N*-acetylmuramic acid and *N*-acetyl-D-glucosamine residues in the peptidoglycan layer (Lowe et al, 1967; Vocadlo et al, 2001). One century after its discovery, numerous studies have demonstrated a great diversity of lysozyme-like proteins that are widespread in all domains of life (Cernooka et al, 2022; Metcalf et al, 2014; Callewaert and Michiels, 2010). Several families of lysozyme-like effectors are substrates of bacterial secretion systems in Gram-negative bacteria (Russell et al, 2012; Callewaert et al, 2012; Sgro et al, 2019). Bacteria produce proteinaceous inhibitors in order to protect the PG from the activity of both endogenous and exogenous hydrolytic enzymes (Callewaert et al, 2012).

The lysozyme-like effector X-Tfe^XAC2609 is a cytoplasmic protein that is transported in a X-T4SS-dependent manner into other bacterial cells. X-Tfi^XAC2610, its cognate inhibitor, has a lipoprotein box motif for localization to the cell periplasm (Souza et al, 2015). We observed that cells expressing a functional X-Tfe^XAC2609 in the absence of X-Tfi^XAC2610 completely abolished bacterial biofilm formation and that biofilms are still not formed by cells with this genetic background when essential X-T4SS components are knocked out. These results show for the first time that inhibition

of endogenous PG hydrolases by immunity proteins can be required for bacterial biofilm formation. Accordingly, another example of the relationship of PG stability and biofilm formation was described in *Campylobacter jejuni*, where PG acetylation is associated with the maintenance of cell wall integrity and contributes to biofilm establishment (Iwata et al, 2016). Also, inhibition of peptidoglycan transpeptidation and transglycosylation by D-leucine and flavomycin can specifically impair biofilm formation in Gram-positive bacteria, although with minimal impact on the bacterial planktonic lifestyle (Bucher et al, 2015).

Two general, not necessarily exclusive, hypotheses were proposed to account for X-Tfe^XAC2609-dependent toxicity in the absence of X-Tfi^XAC2610. The first, which we call *trans*-intoxication or fratricide, hypothesizes that *X. citri* cells inject a cocktail of X-Tfes, including X-Tfe^XAC2609, into neighboring *X. citri* cells and, in the absence of at least one cognate immunity protein, could cause cellular injury and consequent growth suppression or cell death. *Trans*-intoxication has been shown to occur in *P. aeruginosa* cells via the H1-T6SS-dependent secretion of the PG hydrolases Tse1 and Tse3 (Russell et al, 2011). Also, in *Vibrio cholerae*, T6SS-associated immunity proteins have been shown to be important in preventing *trans*-intoxication (Dong et al, 2013). However, in the case of the X-Tfe^XAC2609-dependent toxicity observed in the absence of X-Tfi^XAC2610, a *trans*-intoxication mechanism is not consistent with the observation that both the detrimental effect of X-Tfe^XAC2609 and protection conferred by X-Tfi^XAC2610 are independent of a functional X-T4SS and that wild-type *X. citri* does not outgrow *X. citri* strains lacking X-Tfi^XAC2610 or several other immunity proteins in competition assays (Fig. 2B; Movie EV7). Furthermore, X-Tfe^XAC2609-dependent autolysis in the absence of X-Tfi^XAC2610 does not require the former's C-terminal XVIPCD X-T4SS secretion signal. Thus, we can discard the *trans*-intoxication mechanism or at least propose that this pathway contributes only a small fraction of the X-Tfe^XAC2609-dependent toxic effects observed in the absence of X-Tfi^XAC2610. Nevertheless, the fact that wild-type *X. citri* is unable to kill strains lacking immunity proteins is intriguing. That cells in some way avoid *trans*-intoxication is revealed by the fact that wild-type *X. citri* cells carrying an X-T4SS and full cohort of X-Tfes do not kill the *X. citri* Δ8Δ2609-GFP, the *X. citri* ΔX-Tfe^XAC2609ΔX-Tfi^XAC2610, or any other X-T4SS-deficient strain tested points to a still-to-be-characterized mechanism of protection against *trans*-intoxication (fratricide) that will be addressed in future studies by our group.

We are thus left to consider the *cis*-intoxication hypothesis to explain the toxicity of X-Tfe^XAC2609 and the protection conferred by X-Tfi^XAC2610. Here, the effector exerts its toxic effects within the cell in which it was synthesized. Two cytoplasmic *P. aeruginosa* T6SS effectors, Tse2 and Tse6, have been shown to cause *cis*-intoxication

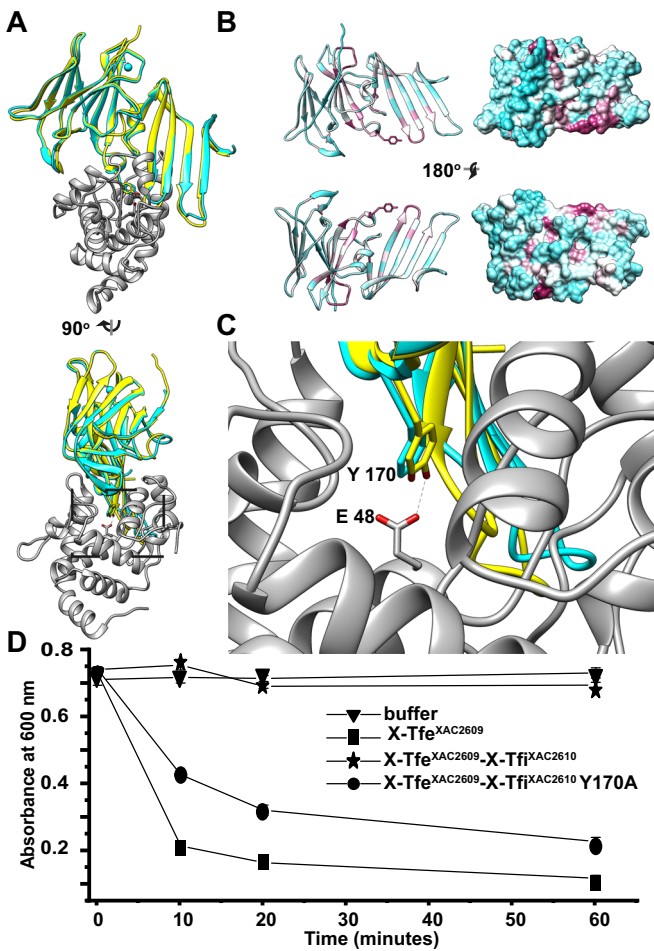

**Figure 6.** Co-evolutionary and structural analysis of the X-Tfe[XAC2609]-X-Tfi[XAC2610] complex by AlphaFold2.

(A) Ribbon representation of the X-Tfe[XAC2609](1–194)-X-Tfi[XAC2610](54–267) complex predicted by AlphaFold2 (Mirdita et al, 2022; Varadi et al, 2022; Jumper et al, 2021). X-Tfe[XAC2609](1–194) is shown in gray and X-Tfi[XAC2610](54–267) is shown in yellow. The X-Tfi[XAC2610](54–267) Alphafold2 model is superposed with the previously determined X-ray crystal structure of X-Tfi[XAC2610] (residues 59–267, cyan; PDB code: 4QTQ) with a bound $Ca^{2+}$ ion (sphere). Root-mean-square deviation (RMSD) of five Alphafold2 models is 0.22 Å, the Predicted Aligned Error (PAE) for the residues and the Predicted Local Distance Difference Test (lDDT) scores (Appendix Fig. S8) also indicate that the models are internally consistent. (B) Cartoon (left) and surface (right) representations of X-Tfi[XAC2610] colored according to degree of conservation (lowest, cyan; highest, purple). (C) Magnification of the protein–protein interface of the model shown in (A) that highlights the interaction between the conserved Y170 of X-Tfi[XAC2610] and E48 of X-Tfe[XAC2609] (both residues shown as sticks). (D) Peptidoglycan hydrolysis assay. *M. luteus* cell wall suspensions were treated with buffer (triangle) or with purified X-Tfe[XAC2609](1–308) (square), X-Tfe[XAC2609](1–308) -X-Tfi[XAC2610](His-55–267) (star), X-Tfe[XAC2609](1–308)-X-Tfi[XAC2610](His-55–267) Y170A and absorbance was monitored at 650 nm. Data points show the mean of $n = 3$ technical replicates +/− SD performed using a single batch of each of the purified proteins. Source data are available online for this figure.

in a T6SS null strain when immunity is depleted (Li et al, 2012; Whitney et al, 2015). In the case of X-Tfe[XAC2609], the toxin somehow makes its way into the cell periplasm where, in the absence of X-Tfi[XAC2610], it degrades the peptidoglycan layer. Analysis of the X-Tfe[XAC2609] sequence by the SignalP 6.0 (Teufel et al, 2022) and other algorithms failed to detect any putative N-terminal signal peptide. Although the mechanism responsible for X-Tfe[XAC2609] transfer into the periplasm is at the moment unknown, we have shown that it is independent of a functional X-T4SS and of the XVIPCD secretion signal. Other proteins have been shown to transfer into the periplasm without any obvious secretion signal, for example VgrG3 from *Vibrio cholerae* (Ho et al, 2017) and heterologously expressed CI2 and HdeA in *E.coli* (Barnes and Pielak, 2011).

We have previously pointed out that the structure of X-Tfi[XAC2610] adopts a similar β-propeller fold topology to two other peptidoglycan hydrolase inhibitors: the *P. aeruginosa* Type VI immunity protein Tsi1 and the *Aeromonas hydrophila* periplasmic i-type lysozyme inhibitor PliI-Ah (Souza et al, 2015). Interestingly, both of these inhibitors block the activity of their respective targets by inserting an exposed loop into the enzyme's active site, in a manner similar to that predicted for the X-Tfe[XAC2609]/X-Tfi[XAC2610] complex (Fig. EV2); this in spite of the fact that they share very low sequence identity with X-Tfi[XAC2610] (7% for Tsi1 and 12% for PliI-Ah)((Souza et al, 2015). In the case of PliI-Ah, its crystal structure in complex with the i-type lysozyme from *Meretrix lusoria* (Ml-iLys) revealed a complementary key-lock interface through the interaction of an exposed loop of PliI-Ah into the substrate-binding groove of Ml-iLys (Herreweghe et al, 2015). Comparison of the topology diagrams of X-Tfi[XAC2610], PliI-Ah and Tsi1 shows that the loops they use to interact with their cognate targets are topologically equivalent (Fig. EV3). While X-Tfi[XAC2610] is predicted to use Y170 to interact directly with E48 in the active site of X-Tfe[XAC2609], in Tsi1 a serine residue (S109) of its inhibitory insertion loop directly interacts with H91 in the active site of the target Tse1 PG amidase enzyme (Benz et al, 2012) (Fig. EV2). In a similar manner, in PliI-Ah, a serine residue (S104) directly recognizes the active site residue E18 of the inhibited lysozyme (Herreweghe et al, 2015) (Fig. EV2).

The structural similarities in the inhibitory mechanisms of X-Tfi[XAC2610], Tsi1 and PliI-Ah are intriguing. While X-Tfi[XAC2610] and PliI-Ah inhibit PG glycohydrolases, Tsi1 inhibits the PG amidase activity of Tse1. It is also worth noting that whereas X-Tfi[XAC2610] inhibits the *cis*-intoxication activity of its cognate effector, the biological function of Tsi1 was described to protect *P. aeruginosa* cells from the deleterious effects of Tse1 molecules transferred from neighboring cells via the H1-T6SS, thus avoiding fratricide or *trans*-intoxication (Russell et al, 2011). Therefore, there are interesting structural and functional relationships among PG hydrolase inhibitors from distinct and diverse biological systems and, even though X-Tfi[XAC2610], Tsi1 and PliI-Ah do not share any detectably relevant sequence identity, they may in fact be very distant members of a homologous protein superfamily that have evolved to inhibit PG hydrolases (such as glycohydrolases and amidases).

## Methods

### Cultivation conditions

The oligonucleotides, plasmids, and bacterial strains used in this work are described in Appendix Tables S3, S4, and S5, respectively. *X. citri* strains were cultivated in LB agar or 2xTY media. The concentrations of the antibiotics used were 70 μg/mL ampicillin,

100 µg/mL spectinomycin, 20 µg/mL gentamicin and 100 µg/mL kanamycin. Experiments involving *X. citri* strain 306 were initiated by picking isolated colonies for inoculation in 5 mL of 2xTY medium supplemented with antibiotics and grown for 12 h. The cells were then harvested, the optical density ($OD_{600nm}$) adjusted to 0.05 with fresh 2xTY medium and a 2 mL volume of this culture was grown at 30 °C, 200 rpm for another 12–18 h.

## Production of *X. citri* gene-knockout strains

Based on the genomic DNA sequence of *X. citri* strain 306 (da Silva et al, 2002), oligonucleotides (Appendix Table S3) were designed to create the desired pNPTS-derived suicide vectors (Appendix Table S4). These vectors were used to produce the *X. citri* simple ΔX-Tfi$^{XAC2610}$, ΔVirB4, ΔVirB5, ΔVirB6, ΔVirB8, and double ΔX-Tfi$^{XAC2610}$/Vir(B(4–10), D4) and ΔX-Tfi$^{XAC2610}$/ΔX-Tfe$^{XAC2609}$ knockout strains using a two-step allelic exchange procedure, as previously described for the ΔVirB7, ΔVirB9, ΔVirB10, and ΔX-Tfe$^{XAC2609}$ strains (Souza et al, 2015, 2011; Sgro et al, 2018). The mutants were confirmed by sequencing, or by colony PCR and western blot.

## Cloning, site-directed mutations, and complementation

From the genomic sequence of *X. citri*, oligonucleotides were designed for complementation of the ΔX-Tfi$^{XAC2610}$ strain (Appendix Table S3). The PCR products were cleaved with restriction enzymes NcoI and SalI, and inserted into the pBRA plasmid vector (M. Marroquin, unpublished). Site-directed mutation was performed using the QuikChange II XL kit (Agilent). Pairs of oligonucleotides F_XAC2609-E48A/R_XAC2609-E48A, F_2610-Y170A/R_2610-Y170A (Appendix Table S3) and plasmids pBRA-2609Nt, pET28a-XAC2610His-22-267 were used to construct the vectors pBRA-XAC2609NtE48A and XAC2610His-22-267, respectively (Appendix Table S4). *X. citri* wild-type and mutant strains were transformed by electroporation (2.0 kV, 25 µF, 200 Ω) and isolated colonies were selected on LB-agar medium containing 100 µg/mL spectinomycin, 70 µg/mL ampicillin.

## Western blotting

After the cultivation of *X. citri* strains carrying pBRA-derived plasmids in 2xTY supplemented with ampicillin and spectinomycin, cells were harvested by centrifugation (5000 rpm, 5 min) and resuspended in water to an $OD_{600nm}$ of 40. A 5-µL aliquot of resuspended cells was lysed in SDS-PAGE loading buffer at 100 °C for 5 min, separated by SDS-PAGE and assayed by western blot. Rabbit polyclonal antibodies serum at 1:1000 dilution anti-X-Tfe$^{XAC2609}$, anti-X-Tfi$^{XAC2610}$ and anti-VirB7 (Souza et al, 2015) were detected by IRDye 800 CW goat anti-rabbit IgG (LI-COR Biosciences) at 1:30,000 dilution. Secondary antibody signals were detected using an Odyssey infrared imaging system (LI-COR Biosciences).

## Colony transparency assay

After the cultivation of *X. citri* strains carrying pBRA-derived plasmids in 2xTY medium supplemented with ampicillin and spectinomycin, cells were harvested by centrifugation (5000 rpm,

5 min), washed in water twice and the $OD_{600nm}$ was normalized to 0.05 in 2xTY medium. Then, 5 µL of each *X. citri* strain culture was applied onto the surface of 1.5% LB-agar plates supplemented with 70 µg/mL ampicillin, spectinomycin and 0.1% arabinose. The plates were cultured at 30 °C and photographs were recorded on a transilluminator every 24 h.

## Colony viability assay

After the cultivation of the *X. citri* inocula in 2xTY supplemented with ampicillin, spectinomycin and 0.1% arabinose, cells were harvested by centrifugation (5000 rpm, 5 min), washed in water twice and the $OD_{600nm}$ was normalized to 0.05. Then, 5 µL of each *X. citri* strains were pipetted onto 1.5% LB-agar plates supplemented with spectinomycin and 0.1% arabinose. Next, the plates were cultured at 30 °C. To assess cell viability, every 24 h of cultivation *X. citri* colonies were removed from the LB-agar plates and gently resuspended in 1 ml of 2xTY. Serial dilutions were made on LB-agar (1.5%) plates supplemented with ampicillin and spectinomycin to estimate viability of colony-forming units per ml (CFU/mL). The results shown are the means of three independent experiments.

## Transmission electron microscopy (TEM)

Liquid cultures of *X. citri* cells 2XTY media were initiated at an $OD_{600nm}$ of 0.05 and grown for 12 h. Cells were collected in microtubes by centrifugation (5000 rpm, 5 min, room temperature), and supernatants were discarded. Cells were washed twice with 1 mL of sodium phosphate buffer solution (0.2 M, pH 7.4). After 15 min at room temperature, cells were collected by centrifugation (6500 rpm, 5 min), and the supernatant was discarded. A fixation step was performed by the addition of 1 mL of modified Karnovsky's solution (2.5% glutaraldehyde, 2% paraformaldehyde, 0.2 M phosphate buffer, pH 7.4) (Watanabe and Yamada, 1983). Then, the samples were incubated for 90 min at room temperature and centrifuged (5000 rpm, 5 min, room temperature), the supernatant was discarded, and an extra washing step was carried out with phosphate buffer as above. A second fixation step was performed by the addition of 1 mL of 1% OsO4 solution followed by incubation for 1 h on ice. The samples were then centrifuged (5000 rpm, 2 min, room temperature) and washed with water. The pellets were dissociated from the microtubes with the use of a glass rod. Thereafter, 1 mL of 0.5% aqueous solution of uranyl acetate was added, followed by another incubation period of 1 h at room temperature. Next, uranyl acetate was removed by centrifugation, and 1 mL of 60% ethanol solution was added. Successive incubations were performed with increasing concentrations of ethanol (70%, 80%, 90%, 95% and 100%). After the last dehydration step with 100% ethanol, 1 mL of propylene oxide solution (Electron Microscopy Sciences) was added, followed by incubation at room temperature for 10 min. Cells were collected by centrifugation as above and three more incubations and washes with propylene oxide were performed. Then, the samples were submitted to four successive exchanges of propylene oxide and resin solutions for microscopy (Low Viscosity embedding Kit (# 14300, Spurr's) in proportions of 1:1, 1:3, 0:1, and 0:1 (propylene oxide:resin) with incubation times of 40 min, 90 min and 12 h and 72 h, respectively. The first 3 incubations (1:1, 1:3, and 0:1) were performed with gentle shaking at room temperature. The fourth incubation step

was performed at 60 °C without agitation in order to solidify the resin. After resin solidification, histological and ultrathin sections were obtained. The slices were collected on 200 mesh carbon-covered copper grids (Ted Pella). Sample imaging was performed on a JEM 2100 JEOL (Central Analitica, Institute of Chemistry, University of São Paulo) and JEOL 100 CX II (Institute of Biomedical Sciences, University of São Paulo) transmission microscopes. Quantitative analysis of the number of cells having a damaged cell envelope and cell counting were performed by manual inspection of the micrography images.

## Time-lapse fluorescence microscopy

Time-lapse fluorescence microscopy assays were carried out as previously described (Oka et al, 2022). Briefly, after the cultivation of the *X. citri* inocula in 2xTY supplemented with the appropriate antibiotics, cells were harvested by centrifugation (5000 rpm, 5 min), washed in water twice and the OD$_{600nm}$ was normalized to 0.5. Then, 1 μL of each *X. citri* strain were pipetted onto a thin LB-agarose support supplemented with propidium iodide (1 μg/mL), appropriate antibiotic and observed with a Nikon Eclipse Ti microscope equipped with filters for GFP (GFP-3035B-000-ZERO, Semrock) and propidium iodide (TxRed-4040B, Semrock) and a Nikon Plan APO 100x objective. Images were collected every 10 min. Image processing and quantitative analysis of the number of cells having a damaged cell envelope and cell counting related to Movies EV1–5 were performed manually using Fiji software (Schindelin et al, 2012) multipoint tool. Time-lapse microscopy was also performed for bacterial competition experiments between *X. citri* wild-type transformed with pBBR-RFP and *X. citri Δ8Δ2609-GFP transformed with pBBR-GFP* (Movie EV6). Cells were grown and washed as above and the cultures were mixed at a 1:1 ratio and microscopy was performed as described above, in the absence of propidium iodide.

## Convolutional Neural Network (CNN) analysis

A pre-trained MobileNet CNN (Howard et al, 2017) was trained using a dataset containing 50 distinct images each of both opaque and transparent *X. citri* colonies (with one or more colonies of a single type per photo). Tensor Flow optimization, data augmentation, and learning rate reduction were employed for training the CNNs. Confidence index for predictions was computed using the np.argmax(prediction) function from the NumPy library (version 1.23.5) (Harris et al, 2020).

## Protein expression and purification

X-Tfi$^{XAC2610}$His-22-267 and X-Tfi$^{XAC2610}$His-22-267_Y170A expression and purification was carried out as previously described (Oka et al, 2022), with some adaptations. Briefly, cells of *E. coli* Bl21 (DE3) carrying the corresponding expression vectors (Appendix Table S4) were grown to OD$_{600nm}$ = 0.8 and the heterologous expression of recombinant proteins was induced using 0.5 mM IPTG for 16 h at 18 °C and 180 rpm in 2xTY medium. After induction of protein expression, cell recovery and lysis, the proteins of interest were submitted to affinity chromatography using HiTrap Ni$^{2+}$-chelating resin (Cytiva) previously equilibrated with 20 mM Tris-HCl buffer (pH 8.0), 200 mM NaCl, 20 mM imidazole,

and 2% (v/v) glycerol, subsequently washed with the same buffer and eluted with an imidazole gradient (20–500 mM). X-Tfe$^{XAC2609}$(1–308) was purified by chromatography as previously described (Souza et al, 2015), using an anion exchange Q-sepharose column (GE Healthcare) and a size exclusion Superdex S200 column (GE Healthcare). Pure proteins were identified by SDS-PAGE, and quantified using absorbance at 280 nm.

## *Micrococcus luteus* peptidoglycan hydrolysis assay

Peptidoglycan hydrolysis experiments were performed as previously described (Souza et al, 2015), with some modifications. Briefly, *M. luteus* cell wall (Sigma) suspensions at OD$_{600nm}$ of 0.7 in 50 mM sodium acetate (pH 5.0) and 2 mM CaCl$_2$ were incubated in triplicate for 60 min at 30 °C with only buffer (negative control), 2 μM X-Tfe$^{XAC2609}$(1–308), 2 μM X-Tfe$^{XAC2609}$(1–308) plus 2 μM X-Tfi$^{XAC2610}$(His-55-267), and 2 μM X-Tfe$^{XAC2609}$(1–308) plus 2 μM of X-Tfi$^{XAC2610}$(His-55–267) Y170A. After 0, 10, 20, and 60 min of incubation, the reactions were quenched by adding 500 mM sodium carbonate and optical density was determined at 600 nm.

## Biofilm formation assay

*X. citri* inocula containing pBRA- and pBBR-derived plasmids were grown for 12 h in 2xTY supplemented with ampicillin, gentamicin, spectinomycin and 0.3% arabinose. After growth, *X. citri* inocula OD$_{600nm}$ was normalized to 0.05 and incubated for 5 days at 30 °C in a laminated microscopy chamber (Nu155411; Lab-Tek, NUNC). Biofilm images were acquired using a Nikon Eclipse Ti microscope equipped with a 100x magnification objective (CFI Plan Apo Lambda 100XH) and a fluorescence filter for GFP (GFP-3035B, Semrock). Images were collected from the base to the top over 20 μm in the *Z* axis at 0.5-μm intervals, and stacked using the FIJI software (Schindelin et al, 2012). The results shown are representative images of three independent experiments. For the 24-well plate-based biofilm assay, 1 mL of *X. citri* inocula were grown for 24 h at 30 °C, 200 rpm in 2xTY medium supplemented with ampicillin, and then allowed to grow at room temperature (22 °C) for 7 days without shaking. To quantify *X. citri* biofilm, a crystal violet-based assay was employed based on that described in (Dunger et al, 2014) with some modifications. *X. citri* cultures were used to inoculate 1 mL 2xTY media in 24-well plates. After 7 days of cultivation without agitation, the liquid medium was carefully removed and replaced with 1 mL of 0.1 M NaCl solution. The biofilm was then resuspended by pipetting and transferred into 1.5-mL microtubes and washed twice with 1 mL of 0.1 M NaCl. Next, the biofilm was resuspended in 1 mL of 0.1% crystal violet solution and incubated for 30 min at room temperature. The biofilm was then collected by centrifugation (6000 rpm, 5 min at 4 °C), the supernatant was discarded, and the insoluble material was washed two more times with 1 mL of 0.1 M NaCl. Finally, the pellets were resuspended by vortexing in 1 mL of 100% ethanol, centrifuged as above and the absorbance of the supernatant was measured at 570 nm.

## Fluorescence spectroscopy

Recombinant X-Tfi$^{XAC2610}$(55–267) was expressed and purified as described above. Fluorescence assays of the purified protein were measured employing an ATF-105 spectrofluorometer (Aviv Biomedical). Thermal denaturation experiments were conducted with 0.5 μM

X-Tfi$^{XAC2610}$(55–267) in 20 mM Tris-HCl (pH 7.5) and 50 mM NaCl. Where indicated, EGTA, MgCl$_2$ or CaCl$_2$ was added to final concentrations of 0.5 mM, 0.75 mM and 0.75 mM, respectively. The temperature was increased between 20 °C and 90 °C, with steps of 2 °C and 6 min equilibration per step. Intrinsic tryptophan fluorescence was excited at 295 nm (bandwidth of 2 nm) and emission was detected at 337 nm (bandwidth of 5 nm). The fraction of folded protein was calculated as described (Pace and Scholtz, 1997).

## Citrus canker assay

Citrus canker assays were performed as previously described (Souza et al, 2011). Briefly, *X. citri* inocula were grown for 12 h in 2xTY supplemented with ampicillin. After growth, *X. citri* cells were diluted in PBS buffer (137 mM NaCl, 2.7 mM KCl, 8 mM Na$_2$HPO$_4$, 2 mM KH$_2$PO$_4$) to an OD$_{600nm}$ of 0.1, and 100 μl used to infiltrate sweet orange leaves (*Citrus sinensis* (L.) Osbeck) using a syringe with a needle. Plants were maintained at 28 °C with a 12 h photoperiod, and symptom development was regularly observed and recorded.

## *X. citri* vs *E. coli* competition assays

Chlorophenol-red β-D-galactopyranoside (CPRG)-based bacterial competition assays were performed as previously described (Oka et al, 2022). Briefly, *X. citri* at OD$_{600nm}$ of 2.0 were mixed 1:1 (v:v) with *E. coli* MG1655 at OD$_{600nm}$ of 11. The mixed cultures (5 μL) were spotted on 1.5% agarose supplemented with 40 μg/mL CPRG (Sigma-Aldrich) in 96 wells plates. Absorbance at 572 nm was monitored every 10 min using a plate reader (SpectraMax Paradigm, Molecular Devices).

## *X. citri* vs *X. citri* competition assays

For the bacterial competition assay based on viability, *X. citri* harboring pBBR5GFP or pBBR2RFP were grown at 30 °C for 12 h in 2xTY supplemented with gentamicin (20 μg/mL) or kanamycin (50 μg/mL), respectively. After three washing steps with fresh 2xTY, the *X. citri* cultures at OD$_{600nm}$ of 2.0 were mixed 1:1 (v:v). Five microliters of the mixture were spotted onto 1% LB-agar plates supplemented with ampicillin (100 μg/mL) and incubated for 40 h at 30 °C. Finally, the colonies were retrieved by resuspension in 1 mL 2xTY medium, and the cellular viability per colony was estimated by serial dilution using selective medium supplemented with kanamycin or gentamicin in LB-agar plates.

## Bioinformatics

Prediction of the structure of the X-Tfi$^{XAC2610}$(54–267)/X-Tfe$^{XAC2609}$(1–194) complex was performed using the UCSF ChimeraX (v1.4, 2022-06-03) software that includes an integrated link to the ColabFold-AlphaFold2 suite (Mirdita et al, 2022; Jumper et al, 2021; Varadi et al, 2022). The search for homologous proteins was performed using the Blast algorithm (Dooley, 2004) with X-Tfi$^{XAC2610}$(54–267) as a query against refseq NCBI protein database (Sayers et al, 2022) using an expect values threshold of $1 \times 10^{-7}$, and outliers and sequences with 99% redundancy were manually discarded. Next, the sequences were realigned with Muscle (Edgar, 2004) using the X-Tfi$^{XAC2610}$ sequence as a reference. The realigned sequence file (Dataset EV1) was used as input for the

Weblogo software (Crooks et al, 2004) to create the conservation profile. Structures were rendered using UCSF Chimera (v1.16) and ChimeraX (v1.4) (Pettersen et al, 2021).

## Data availability

The micrographs used for Figs. 3 and 5A that exceed 300 Mb have been deposited in the Biostudies archive: https://www.ebi.ac.uk/biostudies/bioimages/studies/S-BSST1204.

## Peer review information

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

## Acknowledgements

We thank Dr. Frederico Gueiros-Filho (Instituto de Química, Universidade de São Paulo) for support in the microscopy studies, Central Analítica at Instituto de Química, Universidade de São Paulo and Dr. Ii-Sei Watanabe and Dr. Sonia R Yokomizo from the Instituto de Ciências Biológicas, Universidade de São Paulo for their support and assistance in transmission electron microscopy sample preparation and image acquisition. This work was supported by grants from the Fundação de Amparo à Pesquisa do Estado de São Paulo (FAPESP) to CSF (Projects 2017/17303-7 and 2021/10577-0) and FAPESP fellowships to GUO (Project 2018/09277-9), DPS (2011/50521-1), GGS (2014/04294-1), and GD (2011/22571-4).

## Author contributions

**Gabriel U Oka**: Conceptualization; Data curation; Formal analysis; Validation; Investigation; Methodology; Writing—original draft; Writing—review and editing. **Diorge P Souza**: Conceptualization; Formal analysis; Investigation; Writing—review and editing. **Germán G Sgro**: Investigation; Writing—review and editing. **Cristiane R Guzzo**: Investigation; Writing—review and editing. **German Dunger**: Investigation; Writing—review and editing. **Chuck S Farah**: Conceptualization; Data curation; Formal analysis; Supervision; Funding acquisition; Investigation; Writing—original draft; Project administration; Writing—review and editing.

## Disclosure and competing interests statement

The authors declare no competing interests.

# Expanded View Figures

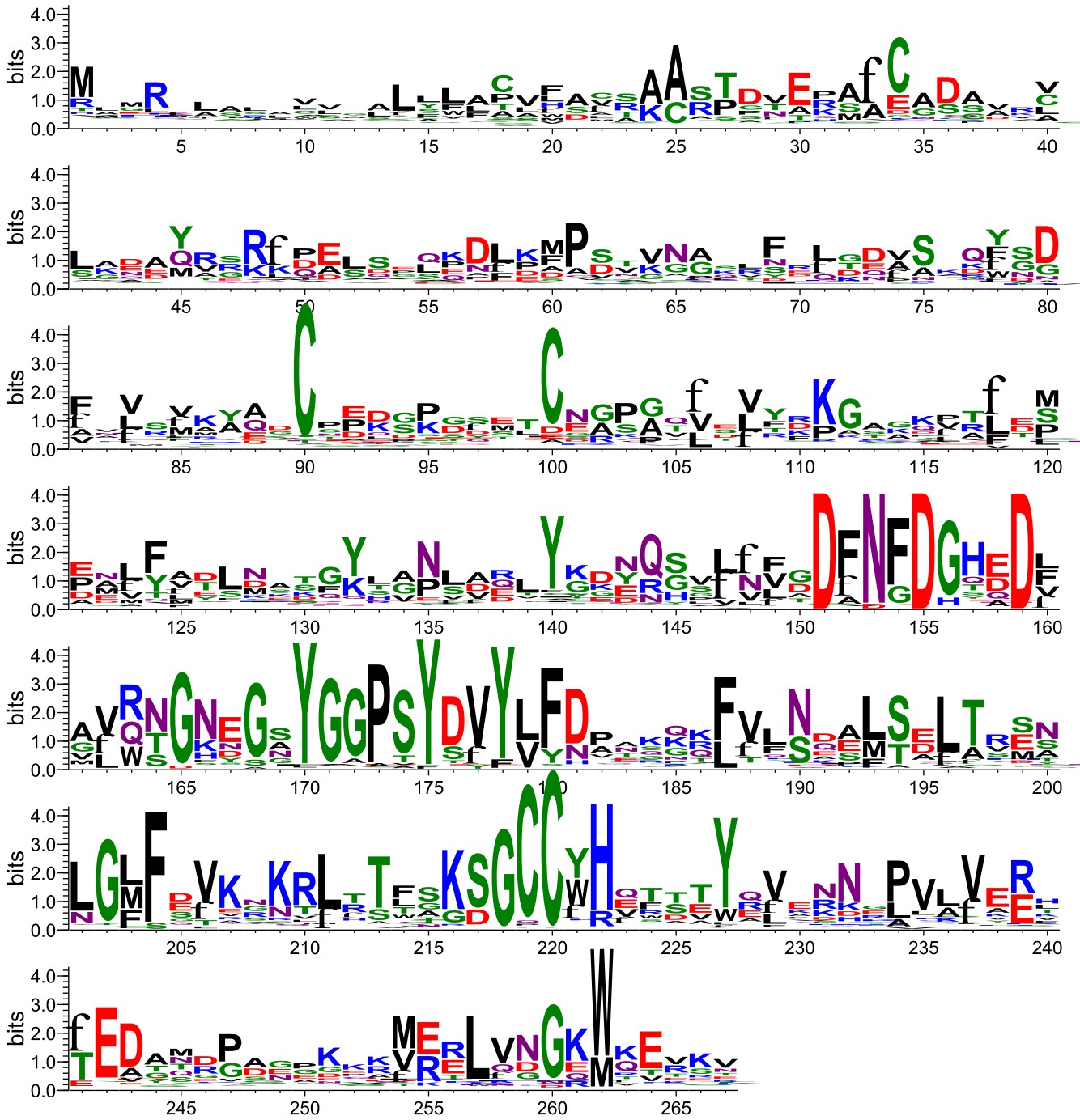

**Figure EV1.  Sequence conservation profile of the multiple alignment of X-Tfi$^{XAC2610}$ homologs.**

The sequences of 429 X-Tfi$^{XAC2610}$ homologs (GenBank code AAM37459/refseq WP_011051709.1) were obtained as described in Material and Methods and are listed in Dataset EV1. The sequence conservation profile was generated using the Weblogo3 server (Crooks et al, 2004). Stacking height indicates sequence conservation at each position while the symbol within the stack indicates the relative frequency of each amino acid in that position. The numbering below the profile corresponds to the amino acid sequence of X-Tfi$^{XAC2610}$. The conserved motif from residues 151–159 corresponds to the Ca$^{2+}$-binding loop observed in the X-Tfi$^{XAC2610}$ crystal structure (Souza et al, 2015). The conserved tyrosine at position 170 is found within a loop predicted to insert into the active site of the cognate effector as described in the main text.

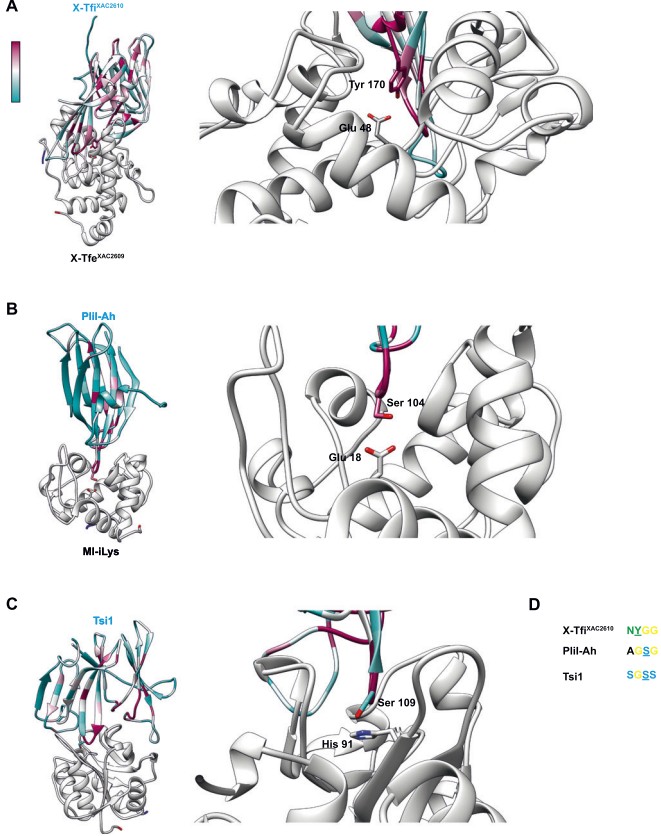

**Figure EV2. Comparison of the model of the X-Tfe<sup>XAC2609</sup>(1–194)-X-Tfi<sup>XAC2610</sup>(54–267) complex with the crystal structures of the PliI-Ah - i-type lysozyme and Tsi1-Tse1 complexes.**

(A–C) Left panels: Ribbon representation of the (A) X-Tfe<sup>XAC2609</sup>-X-Tfi<sup>XAC2610</sup> model generated by Alphafold2. (B) the crystal structure of the periplasmic i-type lysozyme inhibitor from *Aeromonas hydrophila* (PliI-Ah) in complex with the i-type lysozyme from *Meretrix lusoria* (Ml-iLys) (PDB 4PJ2; (Herreweghe et al, 2015) and (C) the crystal structure of Tsi1-Tse1 complex from (PDB 3VPJ) (Ding et al, 2012). (A–C) The inhibitors are colored according to the degree of conservation at each residue position in the corresponding family of homologs (lowest, cyan; highest, purple) and the cognate enzymes are colored in gray. (A–C) Right panels: Zoom of the interaction interfaces showing the insertion of the inhibitory loops into the active sites of the enzymes. Stick models highlight the interactions between residues in the inhibitory loops (X-Tfi<sup>XAC2610</sup> Y170, PliI-Ah S104 and Tsi1 S109) and the enzyme active sites (X-Tfe<sup>XAC2609</sup> E48, Ml-iLys E18, Tse1 H91). (D) Sequences in the inhibitory loops of X-Tfi<sup>XAC2610</sup>, PliI-Ah and Tsi1 that interact directly with the catalytic site of the corresponding toxins. Underlined are X-Tfi<sup>XAC2610</sup> Y170, PliI-Ah S104 and Tsi1 S109.

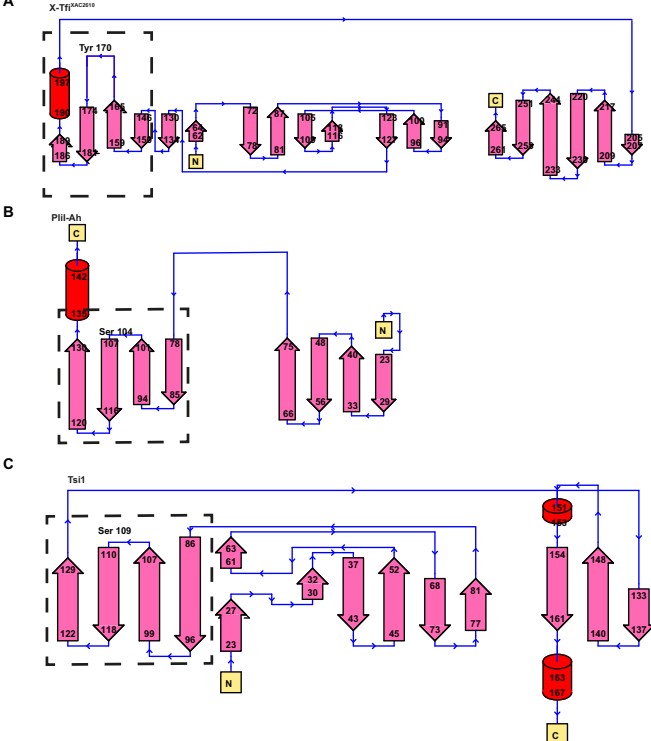

**Figure EV3. Protein topology diagrams of X-Tfi^XAC2610, Pli-Ah and Tsi1.**

Diagrams were generated using the PDBsum server (http://www.ebi.ac.uk/thornton-srv/databases/cgi-bin/pdbsum/GetPage.pl?pdbcode=index.html). Secondary-structure elements are indicated as red cylinders (α-helices) and pink arrows (β-strands). The dotted-squares highlight the common β-sheet found in the three immunity proteins containing a loop between the second and third β-strands that inserts into the active site of the cognate enzyme. (A) X-Tfi^XAC2610 (PDB 4QTQ). (B) Pli-Ah (PDB 4PJ2). (C) Tsi1 (PDB 3VPJ).

