## [Peer Review File · EMBO Reports]

Xanthomonas immunity proteins protect against the cis-toxic effects of their cognate T4SS effectors

Gabriel Oka, Diorge Souza, Germán Sgro, Cristiane Guzzo, German Dunger, and Chuck Farah
DOI: [10.15252/embr.202358138](https://doi.org/10.15252/embr.202358138)

Corresponding author(s): Chuck Farah (chsfarah@iq.usp.br)

Review Timeline:

Transfer from Review Commons:	7th Sep 23
Editorial Decision:	17th Oct 23
Revision Received:	5th Dec 23
Editorial Decision:	14th Dec 23
Revision Received:	22nd Dec 23
Accepted:	8th Jan 24

Editor: Achim Breiling

Transaction Report: This manuscript was transferred to EMBO reports following peer review at Review Commons.

**Review
COMMONS**

Review #1

1. Evidence, reproducibility and clarity:

Evidence, reproducibility and clarity (Required)

The manuscript by Oka et al. shows that one effector of *X. citri* is likely translocated into the periplasm where it cleaves PG unless inhibited by its cognate immunity protein. Interestingly, this effector is required for killing of target cells like *E. coli* in T4SS-dependent manner but it does not seem to be delivered into *X. citri* cells by T4SS. Authors show using various assays that cells lacking the immunity protein have various phenotypes including lysis and defect in biofilm formation, however, despite "cis-intoxication" the ability to kill other bacteria or infect plants remains unaffected. The manuscript is well written and in general all the experiments have proper controls and thus the conclusions seem solid. The results described here are novel and interesting as they are unexpected.

****Major issues that should be addressed:****

- Test various deletion variants of the toxin to identify which part of the protein is responsible for its translocation into the periplasm. This may help to identify the possible mechanism of translocation of the toxin into the periplasm. Alternatively, the authors may attempt to select for non-toxic point mutants of the toxin. This could be done by a random PCR mutagenesis of the toxin and a selection of the surviving mutants in the absence of the immunity protein.
- Test if localization of the immunity protein to the cytoplasm blocks its activity. An immunity protein mutant that lacks its secretion signal should not protect against cis-intoxication.
- While many experiments support the conclusion that the toxin is responsible for "cis-intoxication, the test of "trans-intoxication" should be done again but with the same setup as was used for testing of killing of *E. coli*. The CPRG based assay is far more sensitive than counting survival by plating to count CFUs. This test should be done at a relatively high initial OD so that there is an immediate contact between the "killer" and the "prey" bacteria (lacking immunity/effector). If needed, LacZ should be over-expressed in *X. citri* to make use of the CPRG based assay. In addition, such assay could be used also for "cis-intoxication" to supplement the potentially hard to quantify biofilm experiments shown in Fig. 4 (e.g. test all the T4SS mutants for "cis-intoxication").
- Fig. 2A needs a positive control. For example, test killing of *E. coli* under the same conditions.

- Authors should look at the paper by Ho et al. PNAS 2017, which describes trafficking of VgrG of *V. cholerae* into the periplasm of *E. coli* without an obvious secretion signal. The effector of *X. citri* may behave similarly.
- Provide some form of quantification of the phenotypes (cell rounding and cell death) observed using live-cell imaging.
- Provide quantification of biofilm related phenotypes as well as of the citrus canker development assay.

2. Significance:

Significance (Required)

The study provides an interesting insight into immunity proteins against anti-bacterial toxins. It points to a need to protect against "cis-intoxication". This is novel and interesting to a wide audience of microbiologists interested in bacterial competition as this could be true also for other toxins. It would be however important to identify how is the toxin translocating to the periplasm of the producing bacterium. Some insight into the mechanism would vastly improve the study. My expertise is in understanding bacterial interactions and competition but I lack a direct experience with assays specific for *X. citri*.

3. How much time do you estimate the authors will need to complete the suggested revisions:

Estimated time to Complete Revisions (Required)

(Decision Recommendation)

Between 3 and 6 months

No

Review #2

1. Evidence, reproducibility and clarity:

Evidence, reproducibility and clarity (Required)

This manuscript explores the role of an immunity protein of the *Xanthomonas* type IV secretion system (X-T4SS). In contrast to most T4SSs that conjugate plasmids or transfer effectors into host cells, this system is able to kill other bacteria similar to the role of T6SSs. Here, the authors tested whether the immunity protein XAC2610 functions to prevent cis-intoxication (by self) and/or trans-intoxication (by sister cells). They provide data that the XAC2610 immunity protein functions to protect cis intoxication, but not trans-intoxication, by the T4SS effector XAC2609 (which functions as a peptidoglycan hydrolase). Based on AlphaFold modeling, they went on to identify a residue in XAC2610 that is critical for inhibiting the activity of the XAC2609 toxin. Overall the data is fairly solid and generally support the conclusions the authors made.

****Major comments:****

One of the major conclusions of the manuscript is that XAC2610 does not prevent trans-intoxication and the data in the manuscript support this conclusion. However, I wonder if this is an oversimplification. Notably, the authors observed that wild type Xantho was unable to kill a target cell lacking 8 different toxin/immunity systems (Fig. 1A). One could conclude that none of these immunity proteins function in preventing trans-intoxication ... or ... perhaps it appears that none perform this role because wild-type Xantho never attacks its siblings? For example, it is conceivable that Xantho uses a general mechanism, perhaps somewhat similar to phage exclusion or plasmid incompatibility, to prevent sibling attack? To me this seems more likely than none of the eight immunity proteins play a role in preventing trans-intoxication. Moreover, the phenotype observed for the $\Delta 2610$ mutant in preventing cis-intoxication is somewhat subtle, likely because the toxin and the immunity protein are topologically restricted to the cytoplasm and the periplasm, respectively. This would make sense if this were not the primary role for 2610.

Ideally the authors will be able to test this theory by demonstrating that a wild-type

Xantho strain can attack (but likely not kill) its siblings. Alternatively, could the authors test if related, but not identical, Xantho strains that express 2609/2610 are able to kill their $\Delta 2610$ mutant, i.e. do "cousins" attack each other? Not sure about the semantics but this could be described as preventing trans-intoxication. If they are unable to do either experiment, that is ok but they should at least describe this concept in their discussion (assuming they agree).

Since the cis-intoxication phenotype of the $\Delta 2610$ mutant is subtle, it would strengthen the authors' conclusions on cis-intoxication if they artificially targeted XAC2609 to the periplasm with a sec signal sequence. If the authors are correct, this should be a lethal event in the absence of the 2610 immunity protein. This might be useful in terms of figuring out how the 2609 toxin normally gets into the periplasm, a major unanswered question in this manuscript.

****Minor comments:****

1. Figure 1 is a bit confusing in terms of the layout. It would be beneficial if the authors separated parts A and B by a few spaces.
2. Figure 2A should start off by showing that the Xantho T4SS can kill other bacteria (e.g. Fig S4A). This would set up the paper better.
3. Fig. 2A should include p values.
4. Fig. 2B is really hard to see and should be removed from the manuscript (although I do appreciate the novelty of the technique using Marilyn Monroe).
5. Instead all of the data in Fig. 2B should be shown in a new version of Fig. 2C. Fig. 2C should include additional controls including:
 - a. A wild type strain containing 2609 and 2610 mutants
 - b. A complete virB operon deletion in combination with 2609 and 2610 mutants
 - c. $\Delta 8$ strain
 - d. 2609 lacking its T4SS signal sequence
 - e. 2609 targeted to the periplasm with a sec signal sequence
 - f. etc.
6. Figure 2C. The VirB7 western band looks like in the 2610 complemented strain.
7. Figure 3C should include a comparison of exponential vs. stationary phase cells. In addition, the results for the $\Delta 2610$ mutant and the $\Delta 2610 \Delta B7$ double mutant appear to be different(?). P values should be provided. If it is statistically significant, then this should be explained in the manuscript. It was not clear how the % damaged cells were calculated? # of cells? Stats?
8. The majority of Figure 4 should be replaced by assaying the effect of a virB operon deletion rather than showing the individual mutants.
9. Discussion:

- a. The last one to two paragraphs of the results belong in the Discussion.
- b. A more detailed description of cis-intoxication would be useful.

2. Significance:

Significance (Required)

This work provides a conceptual advance in understanding the protective function of a T4SS immunity protein, X-Tfe XAC2610, against the cis-toxic effects of the T4SS effector, X-Tfi XAC2610. It will likely be of interest to scientists interested in T4SSs & T6SSs and interbacterial competition. Overall this is a thought-provoking manuscript and should be published in a respectable journal.

3. How much time do you estimate the authors will need to complete the suggested revisions:

Estimated time to Complete Revisions (Required)

(Decision Recommendation)

Between 1 and 3 months

No

Review #3

1. Evidence, reproducibility and clarity:

Evidence, reproducibility and clarity (Required)

In this study, the authors suggest that TfeXAC2609-TfiXAC2610 represent a novel deviation from the established paradigm in contact-dependent interbacterial secretion systems. *X. citri* strains lacking the predicted immunity protein, TfiXAC2610, do not suffer a competitive disadvantage when grown in T4SS-inducing conditions against a wild-type strain. Furthermore, cells lacking the immunity develop aberrant morphology and auto-lyse. The mechanism for self-intoxication by TfeXAC2609 is independent of a functional T4SS, and intoxication is exacerbated when the toxin's T4SS-signal sequence is removed.

Major Points

1. The authors of the study do not provide sufficient evidence that TfeXAC2609 contributes to T4SS mediated killing. Does the toxin behave in a synergistic way, rather than mediate killing independently? Does removing the toxin and immunity change the competitive advantage of *X. citri*?

Suggested Experiments: Competing, against *E. coli*, both WT *X. citri* and *X. citri* Δ XAC2609 Δ XAC2610, and determining whether there is a change in relative competitive advantage, or expressing TfeXAC2609 in a heterologous system and marking any observed toxic phenotype.

2. Authors should directly answer where the toxin is active and localized in the cell.

Suggested Experiments: Western blot subcellular fractionation (cytoplasm, periplasm, etc) to determine the localization of each protein.

3. There is no evidence that TfeXAC2609 plays any role in inter-bacterial killing besides that is predicted from its genetic arrangement and in vitro assays from a previous publication.

Suggested Experiments: Again, with the available antibodies, detecting whether TfeXAC2609 is being secreted, either in competition settings against *X. citri* or *E. coli*; given that there is no killing observed in Fig. 2B, it may also be a suitable control for this experiment.

4. The structural and co-evolutionary analysis seems to miss an essential point - that the lack of fratricide protection is not due to a novel protein-protein interaction.

5. The role of the immunity in biofilm formation is confusing. Cells lacking the immunity die within 96 hours (the auto-lysis phenotype). Given that the immunity is required for viability in this time frame, wouldn't it also be required for viability after five days?

Suggested Amendments: Remove or de-emphasize.

6. Why does cell permeability increase with the loss of the T4SS signal sequence?

Without there being greater evidence to support that an alternative secretion system is secreting or transporting the toxin into the periplasm, which may compete with the T4SS, additional hypotheses should be experimentally probed.

7. Unclear if the the loss of cell envelope integrity is a direct effect of TfeXAC2609 activity and not an artifact of cell death. The microscopy also does not show a consistent change in morphology amongst intoxicated cells as there are healthy cells adjacent to lysed cells. This needs to be investigated in much more mechanistic detail.

8. The role for immunity proteins in cis-intoxication is not novel as proposed by the authors. For example, see PMID:22511866 and PMID:26456113 where the authors used an inducible degradation system to show that in a T6SS null strain, cis-intoxication occurs when immunity is depleted.

****Minor Revisions****

1. Inconsistent use of the term "self-killing"; either refers to the killing of kin cells, or self (interchangeably used to refer to trans and cis killing).

2. Terms trans-intoxication and cis-intoxication are convoluted and not constructive to the points being communicated. Self-killing vs kin-killing seem more intuitive and clearer

3. Readability would be improved by the removal of double negatives.

4. Bacterial competition assay in methods only refers to the E. coli competition, not the one between the different genotypes of X. citri.

5. Strain naming scheme presented on pg. 16 doesn't conform to traditional, and clearer, nomenclature typically used.

6. On Pg 25, there is a typo "X-TfiXAC2609" as opposed to X-TfeXAC2609

7. Line 619 - "or several other immunity proteins in competition assays"... where was this data shown? No immediate connection to any figures from this paper nor are there any references.

2. Significance:

Significance (Required)

Overall it is difficult to take paradigm-conflicting conclusions at face-value when they are not presented alongside concrete experimental evidence. Without directly showing that the toxin localizes to the periplasm, the explanation that "the toxin somehow makes its way into the cell periplasm [independent of the T4SS] where it degrades the peptidoglycan layer" hinders the other conclusions presented by the authors.

Consequently, my enthusiasm for this work is minimal.

3. How much time do you estimate the authors will need to complete the suggested revisions:

Estimated time to Complete Revisions (Required)

(Decision Recommendation)

More than 6 months

No

Full Revision

Manuscript number: RC-2023-01856

Corresponding author(s): Chuck S. Farah

[Please use this template only if the submitted manuscript should be considered by the affiliate journal as a full revision in response to the points raised by the reviewers.]

*If you wish to submit a preliminary revision with a revision plan, please use our "Revision Plan" template. **It is important to use the appropriate template to clearly inform the editors of your intentions.**]*

1. General Statements [optional]

This section is optional. Insert here any general statements you wish to make about the goal of the study or about the reviews.

We are happy to resubmit our manuscript "The protective function of an immunity protein against the *cis*-toxic effects of a *Xanthomonas* Type IV Secretion System Effector" by Gabriel Oka et al. This paper shows that the cohort of immunity proteins associated with the cocktail of toxic effectors secreted by the *Xanthomonas citri* T4SS are not required to protect against toxins injected by neighboring cells but rather provide protection against endogenous toxins of the cell in which they were produced. To our knowledge, this the first description of an antibacterial secretion system in which the immunity proteins are dedicated to protecting cells against *cis*-intoxication, a point we emphasize in the revised introduction.

We thank the reviewers for their thorough revision of the manuscript. Two of the three reviewers clearly expressed the opinion that the manuscript would be of general interest and should be published. We have carried out a number of new experiments and data analyses to respond to most of the suggestions of all three reviewers and believe that the manuscript is significantly improved as a result.

This section is mandatory. Please insert a point-by-point reply describing the revisions that were already carried out and included in the transferred manuscript.

Please find below our point-by-point responses to their comments and a description of the revisions that have been incorporated. In the text below, the reviewers' comments are in bold text and our responses are highlighted in yellow.

Reviewer #1 (Evidence, reproducibility and clarity (Required)):

The manuscript by Oka et al. shows that one effector of *X. citri* is likely translocated into the periplasm where it cleaves PG unless inhibited by its cognate immunity protein. Interestingly, this effector is required for killing of target cells like *E. coli* in T4SS-dependent manner but it does not seem to be delivered into *X. citri* cells by T4SS. Authors show using various assays that cells lacking the immunity protein have various phenotypes including lysis and defect in biofilm formation, however, despite "cis-intoxication" the ability to kill other bacteria or infect plants remains unaffected. The manuscript is well written and in general all the experiments have proper controls and thus the conclusions seem solid. The results described here are novel and interesting as they are unexpected.

Major issues that should be addressed:

- Test various deletion variants of the toxin to identify which part of the protein is responsible for its translocation into the periplasm. This may help to identify the possible mechanism of translocation of the toxin into the periplasm. Alternatively, the authors may attempt to select for non-toxic point mutants of the toxin. This could be done by a random PCR mutagenesis of the toxin and a selection of the surviving mutants in the absence of the immunity protein.

Thank you for your insightful experimental suggestions using PCR mutagenesis to investigate the molecular mechanisms of the alternative translocation of X-Tfes to the periplasm. However, I regret to inform you that the first five authors of this manuscript are no longer a member of my lab. Therefore, please consider accepting the results shown in **Figure 5A** where we observe that the N-terminal domain of X-Tfe^{XAC2609} that lacks the XVIPCD domain still abolishes biofilm formation in the absence of X-Tfi^{XAC2610}. Also, note that the E48A point mutation in the active site of the GH19 domain that abolishes the *in vitro* activity of X-Tfe^{XAC2609} (Souza et al. 2015), also abolishes X-TFE^{XAC2609} toxicity *in vivo* in the absence of X-Tfi^{XAC2610} (Figure 5). Furthermore, in the predicted structure of the X-Tfi^{XAC2610}(54-267)-X-Tfe^{XAC2609}(1-194) complex (Figure 6), A tyrosine side chain in the conserved loop in X-Tfi^{XAC2610} interacts directly with Glu48 in the X-Tfe^{XAC2609} active site. One possibility for further investigation is the remaining region of X-Tfe^{XAC2609}(195-306) as a putative translocation domain. Sequence analysis of this region indicates that it encodes a canonical peptidoglycan binding domain. Another possibility is the existing intrinsic leakage of cytoplasmic proteins to the periplasm. As we understand it, the leakage of cytoplasmic proteins to the periplasm is not a well-documented phenomenon, although there is some evidence that suggests it may occur (PMID: 28808000, PMC3016450 references cited in the revised manuscript). This poorly characterized T4SS-independent pathway of translocation as indicated in **Figure 1B (pathway 2)**.

- Test if localization of the immunity protein to the cytoplasm blocks its activity. An immunity protein mutant that lacks its secretion signal should not protect against cis-intoxication.

To address the question, we conducted new assays on colony opacity, as shown in **Figure S2C**. The *X. citri* Δ X-Tfi^{XAC2610} strain was transformed with a plasmid

expressing a cytoplasmic version of X-Tfi^{XAC2610} that lacks the signal peptide and lipobox (X-Tfi^{XAC2610}(His-22-267)). **Figure S2C** shows that this cytosolic version of X-Tfi^{XAC2610} protects *X. citri* from the toxic effects of X-Tfe^{XAC2609} in the Δ X-Tfi^{XAC2610} background. This suggests that X-Tfi^{XAC2610}(His-22-267) may directly interact with X-Tfe^{XAC2609} in the cytoplasm, leading to the inhibition of X-Tfe^{XAC2609} hydrolase activity and/or inhibition of its translocation into the periplasm. This is now mentioned in the results section of the revised manuscript

While many experiments support the conclusion that the toxin is responsible for "cis-intoxication, the test of "trans-intoxication" should be done again but with the same setup as was used for testing of killing of *E. coli*. The CPRG based assay is far more sensitive than counting survival by plating to count CFUs. This test should be done at a relatively high initial OD so that there is an immediate contact between the "killer" and the "prey" bacteria (lacking immunity/effector). If needed, LacZ should be over-expressed in *X. citri* to make use of the CPRG based assay. In addition, such assay could be used also for "cis-intoxication" to supplement the potentially hard to quantify biofilm experiments shown in Fig. 4 (e.g. test all the T4SS mutants for "cis-intoxication").

We are confident that the X-Tfis do not play a role in protecting against T4SS-mediated trans-intoxication since we continue to observe X-Tfe^{XAC2609}-dependent intoxication even in the absence of a functional XT4SS (see experiments using strains lacking X-T4SS subunits in **Figures 2, S2, 3, 4 and 5**). This is not to say that trans-intoxication does not occur. In fact, it does, and there is an independent mechanism that protects against it. We will provide details of the mechanism that protects against trans-intoxication in a forthcoming manuscript. In the present manuscript, we are addressing the phenomenon of *cis*-intoxication. To support our conclusion that the immunity proteins are not involved in the prevention of *trans*-intoxication does not occur in *X. citri*, we have included one additional supplementary video: **Movie S7** shows that wild-type *Xanthomonas citri* does not kill and *X. citri* Δ 8 Δ 2609-GFP. The absence of killing events in these experiments indicates that the X-T4SS-associated X-Tfi immunity proteins are not required for protection against X-T4SS-mediated sibling attack.

In addition, such assay could be used also for "cis-intoxication" to supplement the potentially hard to quantify biofilm experiments shown in Fig. 4 (e.g. test all the T4SS mutants for "cis-intoxication").

- Fig. 2A needs a positive control. For example, test killing of *E. coli* under the same conditions.

Figure 2A of the revised manuscript now shows a CPRG assay competition assay that clearly demonstrates X-T4SS-dependent killing of *E. coli* MG by *X. citri*. We have now included the results of CFU experiments of *X. citri* vs *E. coli* competitions in a new Supplementary Figure (**Figure S1**) that are consistent with the CPRG assays. We note that our group has published similar results in the past (Souza et al. 2015; Oliveira et al. 2016; Oka et al. 2022). CFU measurements of *X. citri* vs *E. coli* competition assays are performed under slightly different conditions from the *X. citri* vs *X. citri* assays shown in **Fig 2B**. This is because *E. coli* grows at a significantly faster rate than *X. citri* so the initial cell ratios in these experiments have to be modified.

Full Revision

- Authors should look at the paper by Ho et al. PNAS 2017, which describes trafficking of VgrG of *V. cholerae* into the periplasm of *E. coli* without an obvious secretion signal. The effector of *X. citri* may behave similarly.

We thank the reviewer for this observation and now mention the paper by Ho et al. in the Discussion of the revised manuscript. Using a number of different algorithms (TatP, SignalP 6.0) we do not find any evidence of putative signal sequences. In the Discussion, we also mention the manuscript by Dong et al., 2013 that showed that the immunity protein TsiV3 that neutralizes VgrG3 is critical to prevent trans-intoxication.

- Provide some form of quantification of the phenotypes (cell rounding and cell death) observed using live-cell imaging.

As suggested by the reviewer, we performed a quantitative analysis of the propidium iodide (PI) permeability by calculating the percentage of PI permeable cells observed in movies S1-S5. This data is now presented in **Figure 3** and **Table S4** of the revised manuscript.

- Provide quantification of biofilm related phenotypes as well as of the citrus canker development assay

As suggested by the reviewer, we have carried out experiments to quantify the amount of biofilm using a crystal violet assay (absorbance at 570 nm). The results are presented in **Figure S5** of the revised manuscript.

Reviewer #1 (Significance (Required)):

The study provides an interesting insight into immunity proteins against anti-bacterial toxins. It points to a need to protect against "cis-intoxication". This is novel and interesting to a wide audience of microbiologists interested in bacterial competition as this could be true also for other toxins.

We thank the reviewer for his/her positive recommendation.

It would be however important to identify how is the toxin translocating to the periplasm of the producing bacterium. Some insight into the mechanism would vastly improve the study. My expertise is in understanding bacterial interactions and competition but I lack a direct experience with assays specific for *X. citri*.

We agree that an understanding of the mechanism of translocation into the periplasm would be interesting but is beyond the scope of the present manuscript. However, we do point out that this has been observed previously by other groups in the fourth paragraph of the Discussion of the revised manuscript: "... In the case of X-Tfe^{XAC2609}, the toxin somehow makes its way into the cell periplasm where, in the absence of X-Tfi^{XAC2610}, it degrades the peptidoglycan layer. Analysis of the X-Tfe^{XAC2609} sequence by the SignalP 6.0 (Teufel et al., 2022) and other algorithms failed to detect any putative N-terminal signal peptide. Although the mechanism responsible for X-Tfe^{XAC2609} transfer into the periplasm is at the moment unknown, we have shown that it is

independent of a functional X-T4SS and of the XVIPCD secretion signal. Other bacterial proteins have been shown to transfer into the periplasm without any obvious secretion signal, for example VgrG3 from *Vibrio cholerae* (Ho et al. 2017) and recombinant forms of HdeA and chymotrypsin inhibitor 2 (Banes and Pielak, 2011)."

Reviewer #2 (Evidence, reproducibility and clarity (Required)):

This manuscript explores the role of an immunity protein of the *Xanthomonas* type IV secretion system (X-T4SS). In contrast to most T4SSs that conjugate plasmids or transfer effectors into host cells, this system is able to kill other bacteria similar to the role of T6SSs. Here, the authors tested whether the immunity protein XAC2610 functions to prevent cis-intoxication (by self) and/or trans-intoxication (by sister cells). They provide data that the XAC2610 immunity protein functions to protect cis intoxication, but not trans-intoxication, by the T4SS effector XAC2609 (which functions as a peptidoglycan hydrolase). Based on AlphaFold modeling, they went on to identify a residue in XAC2610 that is critical for inhibiting the activity of the XAC2609 toxin. Overall the data is fairly solid and generally support the conclusions the authors made.

Major comments:

One of the major conclusions of the manuscript is that XAC2610 does not prevent trans-intoxication and the data in the manuscript support this conclusion. However, I wonder if this is an oversimplification. Notably, the authors observed that wild type Xantho was unable to kill a target cell lacking 8 different toxin/immunity systems (Fig. 1A). One could conclude that none of these immunity proteins function in preventing trans-intoxication ... or ... perhaps it appears that none perform this role because wild-type Xantho never attacks its siblings? For example, it is conceivable that Xantho uses a general mechanism, perhaps somewhat similar to phage exclusion or plasmid incompatibility, to prevent sibling attack? To me this seems more likely than none of the eight immunity proteins play a role in preventing trans-intoxication. Moreover, the phenotype observed for the $\Delta 2610$ mutant in preventing cis-intoxication is somewhat subtle, likely because the toxin and the immunity protein are topologically restricted to the cytoplasm and the periplasm, respectively. This would make sense if this were not the primary role for 2610.

Ideally the authors will be able to test this theory by demonstrating that a wild-type Xantho strain can attack (but likely not kill) its siblings. Alternatively, could the authors test if related, but not identical, Xantho strains that express 2609/2610 are able to kill their $\Delta 2610$ mutant, i.e. do "cousins" attack each other? Not sure about the semantics but this could be described as preventing trans-intoxication. If they are unable to do either experiment, that is ok but they should at least describe this concept in their discussion (assuming they agree).

We thank the reviewer for his insightful comments. Indeed, this manuscript is focussed solely on the role of X-Tfi immunity proteins which we show to be principally involved in avoiding cis-intoxication (self-intoxication). The question of trans-intoxication will be left to an upcoming manuscript by our group. In fact we have identified a key factor (not an X-Tfi) that is responsible

for inhibiting *trans*-intoxication. As suggested by the reviewer, we have now added the following text to the end of the third paragraph of the Discussion: “Nevertheless, the fact that wild-type *X. citri* is unable to kill strains lacking immunity proteins is intriguing. That cells in some way avoid *trans*-intoxication is revealed by the fact that *X. citri* wild-type cells carrying an X-T4SS and full cohort of X-Tfes do not kill the *X. citri* $\Delta 8\Delta 2609$ -GFP, the *X. citri* ΔX -Tfe^{XAC2609} ΔX -Tfi^{XAC2610}, or any other X-T4SS-deficient strain tested points to a still-to-be-characterized mechanism of protection against *trans*-intoxication (fratricide) that will be addressed in future studies by our group.”

Minor comments:

1. **Figure 1 is a bit confusing in terms of the layout. It would be beneficial if the authors separated parts A and B by a few spaces.**

As suggested, we have modified the layout of **Figure 1** to more clearly distinguish between the two mechanisms tested.

2. **Figure 2A should start off by showing that the Xantho T4SS can kill other bacteria (e.g. Fig S4A). This would set up the paper better.**

As suggested, old Figure S4A has now been transferred to **Figure 2A** in the revised manuscript.

3. **Fig. 2A should include p values.**

As suggested, p values have been provided in the legend of **Figure 2B** (old Figure 2A).

4. **Fig. 2B is really hard to see and should be removed from the manuscript (although I do appreciate the novelty of the technique using Marilyn Monroe).**

We have transferred the old Fig. 2B to the Supplementary Material (**Fig S2**) of the revised manuscript. We agree that the effect is subtle, but we want to maintain the figure since the transparency of the $\Delta XAC2610$ *X. citri* colonies over time were the first observations that led us to investigate this phenomenon. Additionally, to reduce potential human bias and to enhance the objectivity of the assay, we employed a Convolutional Neural Network (CNN) to analyze all the colonies presented in **Fig S2**. This method provides a confidence tendency index for opacity and transparency variations. A detailed description of this new methodology is in the "Materials and Methods" section (Convolutional Neural Network (CNN) analysis).

5. **Instead all of the data in Fig. 2B should be shown in a new version of Fig. 2C. Fig. 2C should include additional controls including:**

- a. **A wild type strain containing 2609 and 2610 mutants**
- b. **A complete virB operon deletion in combination with 2609 and 2610 mutants**

Full Revision

- c. $\Delta 8$ strain
- d. 2609 lacking its T4SS signal sequence
- e. 2609 targeted to the periplasm with a sec signal sequence
- f. etc.

We sincerely value the comprehensive suggestions for improving what was previously presented as Fig. 2D. (Current version of Figure 2B is the Fig S2 as mentioned in the previous observation). We encounter a practical challenge here: the primary authors responsible for these experiments, especially the first five, have since departed from our lab. This situation limits our immediate capacity to execute the extensive set of experiments you've proposed.

Recognizing the significance of the controls you've outlined for a quantitative analysis of the colony phenotypes (**Fig. 2C** (current version)) we have instead supplemented our study with a rigorous quantitative analysis of the microscopy assays referenced in **Movies S1-S5**, **Figure 3**, **Table S4**. These analyses further emphasize our observations concerning colony transparency (**Fig S2**).

6. Figure 2C. The VirB7 western band looks like in the 2610 complemented strain.

Thank you for pointing out the discrepancy in our previous manuscript at line 366, which pertains to the description of the mutants in old **Fig. 2**. The double mutant, $\Delta X\text{-Tfi}^{\text{XAC2610}}\Delta\text{virB7}$ strain, was actually complemented with $X\text{-Tfi}^{\text{XAC2610}}$ (as stated in the current version (**Fig S2B**)), and not with VirB7. Additionally, we have corrected the legend of the figure (line 684 previous version) from $(\Delta X\text{-Tfi}^{\text{XAC2610}}\Delta\text{VirB7c})$ to $(\Delta X\text{-Tfi}^{\text{XAC2610}}\Delta\text{VirB7})$. We apologize for the mix-up in our earlier description and are grateful for your meticulous review and feedback in this matter. Furthermore, we agree that, in this particular experiment, the VirB7 band seems weaker but it is clearly visible in the 2610 complemented strain.

7. Figure 3C should include a comparison of exponential vs. stationary phase cells. In addition, the results for the $\Delta 2610$ mutant and the $\Delta 2610 \Delta B7$ double mutant appear to be different(?). P values should be provided. If it is statistically significant, then this should be explained in the manuscript. It was not clear how the % damaged cells were calculated? # of cells? Stats?

The statistical analysis that the reviewer suggested has been provided in the new version of the **Figure 4C** and its legend. In addition, we have also included a supplementary **Table S5** that presents the total number of cells analyzed in these experiments.

Full Revision

8. The majority of Figure 4 should be replaced by assaying the effect of a *virB* operon deletion rather than showing the individual mutants.

We believe that retaining old Figure 4 (Figure 5 of the revised manuscript) is important. By showcasing results from this specific set of single mutants, we are able to rule out the possibility that X-Tfe^{XAC2609} translocation into the periplasm is mediated by a distinct X-T4SS subunit or subcomplex. We've expanded on this rationale at the start of the paragraph to provide a more comprehensive justification for our approach.

9. Discussion:

- a. The last one to two paragraphs of the results belong in the Discussion.
- b. A more detailed description of *cis*-intoxication would be useful.

As suggested, the last two paragraphs the Results section of the original manuscript have now been moved to the end of the Discussion.

As suggested by the reviewer, third paragraph of the Discussion describes *cis*-intoxication in more detail.

Reviewer #2 (Significance (Required)):

This work provides a conceptual advance in understanding the protective function of a T4SS immunity protein, X-Tfe XAC2610, against the *cis*-toxic effects of the T4SS effector, X-Tfi XAC2610. It will likely be of interest to scientists interested in T4SSs & T6SSs and interbacterial competition. Overall this is a thought-provoking manuscript and should be published in a respectable journal.

We sincerely thank Reviewer #2 for the thoughtful appraisal and positive feedback regarding our work. We are gratified to hear that the reviewer recognizes the conceptual advance our research brings to the understanding of T4SS immunity proteins and are encouraged by the acknowledgment that this manuscript will be of interest to our peers. We truly appreciate the endorsement for publication in a reputable journal.

Reviewer #3 (Evidence, reproducibility and clarity (Required)):

In this study, the authors suggest that TfeXAC2609-TfiXAC2610 represent a novel deviation from the established paradigm in contact-dependent interbacterial secretion systems. *X. citri* strains lacking the predicted immunity protein, TfiXAC2610, do not suffer a competitive disadvantage when grown in T4SS-inducing conditions against a wild-type strain. Furthermore, cells lacking the immunity develop aberrant morphology and auto-lyse. The mechanism for self-intoxication by Tfe^{XAC2609} is independent of a

functional T4SS, and intoxication is exacerbated when the toxin's T4SS-signal sequence is removed.

Major Points

1. The authors of the study do not provide sufficient evidence that TfeXAC2609 contributes to T4SS mediated killing. Does the toxin behave in a synergistic way, rather than mediate killing independently? Does removing the toxin and immunity change the competitive advantage of *X. citri*?

We have shown in a previous publication that X-Tfe^{XAC2609} does contribute to X-T4SS mediated killing (Oka et al, 2022). In that published paper we show that even in the absence of seven other toxin/antitoxin pairs, X-T4SS mediated transfer of only one effector (X-Tfe^{XAC2609} or X-Tfe^{XAC3634}) can kill *E. coli* cells.

Removing only X-Tfe^{XAC2609} and X-Tfi^{XAC2610} does not significantly reduce the ability of *X. citri* cells to kill *E. coli* (Fig. 2A of the revised manuscript). This is expected since this double mutant still retains seven other toxin/immunity pairs.

Suggested Experiments: Competing, against *E. coli*, both WT *X. citri* and *X. citri* Δ XAC2609 Δ XAC2610, and determining whether there is a change in relative competitive advantage, or expressing TfeXAC2609 in a heterologous system and marking any observed toxic phenotype.

The results of the experiment suggested by the reviewer have now been included in part A of the revised version of Figure 2. The effect of deleting only one toxin such as X-Tfe^{XAC2609} results in no detectable difference in killing efficiency, most likely due to the presence of the eight other X-Tfes, three of which have been shown (XAC3634) or are predicted (XAC0466 and XAC1918) to have peptidoglycan hydrolase activity (Oka et al, 2022, Souza, 2015, Sgro et al, 2019).

2. Authors should directly answer where the toxin is active and localized in the cell.

Suggested Experiments: Western blot subcellular fractionation (cytoplasm, periplasm, etc) to determine the localization of each protein.

In response to the query about the toxin's activity and localization within the cell, we acknowledge the importance of such experiments to shed light on these aspects. However, I would like to highlight that the five first authors of this work are no longer affiliated with our lab. Consequently, we are facing constraints in terms of manpower and expertise to undertake comprehensive experiments such as the suggested subcellular fractionation.

Also, our earlier work demonstrated the importance of the XVIPCD for secretion via X-T4SS (Souza, 2015) and *in vivo* activity of X-Tfe^{XAC2609} (Oka et al., 2022). Moreover, using heterologous proteins expressed in *E. coli* (Souza, 2015) and our current

observation that the absence of X-TfiXAC2610 induces spheroplast formation (**Fig 4A-B, Movie S6**) strongly suggest that the peptidoglycan glycohydrolase activity of the N-terminal domain of X-Tfe^{XAC2609} acts in the periplasm.

3. There is no evidence that TfeXAC2609 plays any role in inter-bacterial killing besides that is predicted from its genetic arrangement and in vitro assays from a previous publication.

Suggested Experiments: Again, with the available antibodies, detecting whether TfeXAC2609 is being secreted, either in competition settings against *X. citri* or *E. coli*; given that there is no killing observed in Fig. 2B, it may also be a suitable control for this experiment.

We have published *in vivo* evidence in the past:

Souza et al, 2015 showed that X-Tfe^{XAC2609} is secreted when in contact with *E. coli* cells.

Oka et al, 2022 showed that an *X. citri* strain expressing X-Tfe^{XAC2609}/X-Tfi^{XAC2610} but lacking seven other toxin/antitoxin pairs can still kill *E. coli*.

4. The structural and co-evolutionary analysis seems to miss an essential point - that the lack of fratricide protection is not due to a novel protein-protein interaction.

We do not understand this comment. As we point out in the manuscript, X-Tfi^{XAC2610} does not protect against fratricide (*trans*-intoxication) but instead does protect against suicide (*cis*-intoxication). This protection requires a X-Tfe^{XAC2609}-X-Tfi^{XAC2610} protein-protein interaction supported by the structural and co-evolutionary analysis as well as the experimental data using the X-Tfi^{XAC2610} Y170A mutant (**Fig. 6D** of the revised manuscript). Moreover, we believe that the structural and sequence analysis significantly expand the knowledge of the broader family of immunity proteins to which X-TfiXAC2610 belongs (**Fig. S10** and **Fig. S11** of the revised manuscript).

5. The role of the immunity in biofilm formation is confusing. Cells lacking the immunity die within 96 hours (the auto-lysis phenotype). Given that the immunity is required for viability in this time frame, wouldn't it also be required for viability after five days?

Suggested Amendments: Remove or de-emphasize.

In the manuscript we use several different techniques to show that cells lacking the X-Tfi^{XAC2610} immunity protein are less viable than the wild-type strain under certain conditions (growth on LB agar plates, biofilm formation) but perhaps not under others (ie in direct short-term competition experiments against *E. coli* and in long-term (2 week) *in planta* citrus canker assays). This is consistent with the fact that ultrastructural analysis by transmission electron microscopy shows that when grown in liquid media, only around 2% of *X. citri* cells lacking X-Tfi^{XAC2610} present significant damage to their cell envelope (only 0.1% of wild-type cells show damage).

6. Why does cell permeability increase with the loss of the T4SS signal sequence?

Without there being greater evidence to support that an alternative secretion system is secreting or transporting the toxin into the periplasm, which may compete with the T4SS, additional hypotheses should be experimentally probed.

The reviewer is comparing the propidium iodide permeability results observed for the $\Delta X\text{-Tfi}^{\text{XAC2610}}$ mutant (carrying an empty pBRA plasmid) that expresses full-length X-Tfe^{XAC2609} from its chromosomal gene with the $\Delta X\text{-Tfe}^{\text{XAC2609}}/\Delta X\text{-Tfi}^{\text{XAC2610}}$ double mutant carrying the pBRA-X-Tfe^{XAC2609NT} plasmid that expresses the X-Tfe^{XAC2609} protein lacking the T4SS signal sequence from a very strong inducible promoter. Therefore, it can be expected that the levels of the truncated effector could be significantly greater than that of the full-length effector, leading to more damage.

Note that, in the absence of X-Tfi^{XAC2610}, cell permeability increases only if X-Tfe^{XAC2609} is present, with or without its XVIPCD T4SS signal sequence. This is consistent with a *cis*-intoxication mechanism which is independent of the X-T4SS-mediated transfer of the toxin from one cell to another. As we mention in the revised manuscript, and as pointed out by reviewer 1, Ho et al have also observed that when a lysozyme-containing domain of the T6SS effector VgrG3 is expressed in *E. coli* or in *Vibrio cholerae*, it can be detected in the periplasm in spite of the lack of a detectable signal sequence and in the absence of a functional T6SS. Ho et al attributed this observation to a “cryptic” secretion mechanism.

7. Unclear if the the loss of cell envelope integrity is a direct effect of TfeXAC2609 activity and not an artifact of cell death. The microscopy also does not show a consistent change in morphology amongst intoxicated cells as there are healthy cells adjacent to lysed cells. This needs to be investigated in much more mechanistic detail.

We observed that the X-Tfe^{XAC2609} toxicity is dependent on its lysozyme domain since a point mutation in the active site residue (E48A) abolishes the toxicity-related phenotype in the biofilm assay (Figure 5).

8. The role for immunity proteins in *cis*-intoxication is not novel as proposed by the authors. For example, see PMID:22511866 and PMID:26456113 where the authors used an inducible degradation system to show that in a T6SS null strain, *cis*-intoxication occurs when immunity is depleted.

We thank the reviewer for pointing out these observations which are now mentioned and cited in the Introduction and in the Discussion of the revised manuscript.

Minor Revisions

1. Inconsistent use of the term "self-killing"; either refers to the killing of kin cells, or self (interchangeably used to refer to trans and cis killing).

The term “self-killing” no longer appears in the manuscript.

2. Terms trans-intoxication and cis-intoxication are convoluted and not constructive to the points being communicated. Self-killing vs kin-killing seem more intuitive and clearer. We prefer to maintain the use of the terms *cis*-intoxication and *trans*-intoxication which we

defined in the Introduction, at the beginning of the Results section and in the Discussion as well as in **Figure 1**.

3. Readability would be improved by the removal of double negatives.

We have tried to avoid these whenever possible.

4. Bacterial competition assay in methods only refers to the *E. coli* competition, not the one between the different genotypes of *X. citri*.

Both methods were described in the same paragraph in the original manuscript. For clarity, this has now been divided into two sections in the revised manuscript: "*X. citri* vs *E. coli* competition assays" and "*X. citri* vs *X. citri* competition assays".

5. Strain naming scheme presented on pg. 16 doesn't conform to traditional, and clearer, nomenclature typically used.

We have checked the manuscript to make sure that strain naming was consistent throughout the manuscript.

6. On Pg 25, there is a typo "X-TfiXAC2609" as opposed to X-TfeXAC2609

Thank you for the observation. This has now been corrected.

7. Line 619 - "or several other immunity proteins in competition assays"... where was this data shown? No immediate connection to any figures from this paper nor are there any references.

This is shown in **Figure 2B** and in **Movie S7** which is now cited directly in the revised manuscript.

Reviewer #3 (Significance (Required)):

Overall it is difficult to take paradigm-conflicting conclusions at face-value when they are not presented alongside concrete experimental evidence. Without directly showing that the toxin localizes to the periplasm, the explanation that "the toxin somehow makes its way into the cell periplasm [independent of the T4SS] where it degrades the peptidoglycan layer" hinders the other conclusions presented by the authors. Consequently, my enthusiasm for this work is minimal.

We deeply appreciate the insightful feedback from Reviewer #3, particularly regarding the concerns about paradigm-conflicting conclusions. We are steadfast in our commitment to ensuring that our findings are both rigorous and scientifically relevant.

Evidence for Toxin Localization: We understand the criticality of concrete experimental evidence for toxin localization to the periplasm. While our data suggest an yet to be discovered translocation pathway of X-Tfe^{XAC2609} from the cytoplasm to the periplasm, we recognize the importance of providing direct evidence. We are actively working on methodologies to understand this phenomenon. However, we do not believe that answering this question is absolutely necessary to understand the main conclusions of the present manuscript.

Dear Dr. Farah,

Thank you for the transfer of your revised manuscript to our editorial offices. I have now received the reports from the three referees that were asked to re-evaluate your study, you will find below. As you will see, the referees now support publication of your study in EMBO reports. Referee #3 (referee #2 from the Review Commons submission) has some minor remaining concerns and suggestions to improve the manuscript I ask you to address in a final revised version. Please also provide a final p-b-p-response addressing these points.

Moreover, the manuscript needs formatting according to our journal style. Please carefully review the instructions that follow below.

When submitting your final revised manuscript, we will require:

1) a .docx formatted version of the final manuscript text (including legends for main figures, EV figures and tables), but without the figures included. Figure legends should be compiled at the end of the manuscript text.

2) individual production quality figure files as .eps, .tif, .jpg (one file per figure), of main figures and EV figures. Please upload these as separate, individual files upon re-submission.

The Expanded View format, which will be displayed in the main HTML of the paper in a collapsible format, has replaced the Supplementary information. You can submit up to 5 images as Expanded View. Please follow the nomenclature Figure EV1, Figure EV2 etc. The figure legend for these should be included in the main manuscript document file in a section called Expanded View Figure Legends after the main Figure Legends section. Additional Supplementary material should be supplied as a single pdf file labeled Appendix. The Appendix should have page numbers and needs to include a table of content on the first page (with page numbers) and legends for all content. Please follow the nomenclature Appendix Figure Sx, Appendix Table Sx etc. throughout the text, and also label the figures and tables according to this nomenclature. In this case, I think it will be possible to combine some of the Supplementary Figures to have in the end 5 EV figures. It is not necessary to have one EV figure related to one main figure. Just make sure that the panels are called out correctly.

3) a complete author checklist, which you can download from our author guidelines (<https://www.embopress.org/page/journal/14693178/authorguide>). Please insert page numbers in the checklist to indicate where the requested information can be found in the manuscript. The completed author checklist will also be part of the RPF.

4) that primary datasets produced in this study (e.g. RNA-seq, ChIP-seq, structural and array data) are deposited in an appropriate public database. If no primary datasets have been deposited, please also state this in a dedicated section (e.g. 'No primary datasets have been generated and deposited'), see below.

The accession numbers and database should be listed in a formal "Data Availability" section (placed after Materials & Methods) that follows the model below. This is now mandatory (like the COI statement). Please note that the Data Availability Section is restricted to new primary data that are part of this study. This section is mandatory. As indicated above, if no primary datasets have been deposited, please state this in this section

Data availability

5) We now request the publication of original source data with the aim of making primary data more accessible and transparent to the reader. It seems our source data coordinator has already contacted you and informed you which figure panels we would need source data for and how to upload and organize the files. Please do that during re-submission of the manuscript files and also upload the filled in source data checklist (attached).

6) Our journal encourages inclusion of *data citations in the reference list* to directly cite datasets that were re-used and obtained from public databases. Data citations in the article text are distinct from normal bibliographical citations and should directly link to the database records from which the data can be accessed. In the main text, data citations are formatted as follows: "Data ref: Smith et al, 2001" or "Data ref: NCBI Sequence Read Archive PRJNA342805, 2017". In the Reference list, data citations must be labeled with "[DATASET]". A data reference must provide the database name, accession number/identifiers and a resolvable link to the landing page from which the data can be accessed at the end of the reference. Further instructions are available at: <http://www.embopress.org/page/journal/14693178/authorguide#referencesformat>

7) Regarding data quantification and statistics, please make sure that the number "n" for how many independent experiments were performed, their nature (biological versus technical replicates), the bars and error bars (e.g. SEM, SD) and the test used to calculate p-values is indicated in the respective figure legends (also for potential EV and Appendix figures). Please also check that all the p-values are explained in the legend, and that these fit to those shown in the figure. Please provide statistical testing where applicable. Please avoid the phrase 'independent experiment', but clearly state if these were biological or technical replicates. Please also indicate (e.g. with n.s.) if testing was performed, but the differences are not significant. In case n=2, please show the data as separate datapoints without error bars and statistics. See also: <http://www.embopress.org/page/journal/14693178/authorguide#statisticalanalysis>

8) Please add scale bars of similar style and thickness to all the microscopic images, using clearly visible black or white bars (depending on the background). Please place these in the lower right corner of the images themselves. Please do not write on or near the bars in the image but define the size in the respective figure legend.

9) Please also note our reference format:

10) We updated our journal's competing interests policy in January 2022 and request authors to consider both actual and perceived competing interests. Please review the policy <https://www.embopress.org/competing-interests> and update your competing interests if necessary. Please name this section 'Disclosure and Competing Interests Statement' and put it after the Acknowledgements section.

11) We now use CRediT to specify the contributions of each author in the journal submission system. CRediT replaces the author contribution section. Please use the free text box to provide more detailed descriptions and do NOT add a author contributions section to the manuscript text file. See also guide to authors: <https://www.embopress.org/page/journal/14693178/authorguide#authorshippinguidelines>

12) Please add up to 5 keywords to the manuscript and order the manuscript sections like this, using these names: Title page - Abstract - Keywords - Introduction - Results - Discussion - Materials and Methods - Data availability section - Acknowledgements - Disclosure and Competing Interests Statement - References - Figure legends - Expanded View Figure legends - Tables

13) Please provide a final title with not more than 100 characters (including spaces).

14) Please enter all the funding information also into our submission system and make sure this is complete and similar to the one mentioned in the manuscript text file.

15) Please provide a final abstract with not more than 175 words.

In addition, I would need from you:

- a short, two-sentence summary of the manuscript (not more than 35 words).
- three to four short (!) one sentence bullet points highlighting the key findings of your study.
- a schematic summary figure (synopsis image) in jpeg or tiff format with the exact width of 550 pixels and a height of not more than 400 pixels that can be used as a visual synopsis on our website.

I look forward to seeing revised version of your manuscript when it is ready. Please let me know if you have questions or comments regarding the revision.

Best,

Referee #1:

Thank you for addressing all my concerns in the revised manuscript.

Referee #2 (referee #3 RC):

While it is disappointing that the authors were unable to perform the requested sub-cellular localization experiment, they have satisfactorily addressed my other concerns in their revised manuscript.

Referee #3 (referee #2 RC):

Sorry for the delay in submitting my review.

Overall I am pleased with the modifications and think the manuscript is close to being accepted in EMBORep with a few minor changes.

1. Unfortunately, I still think the manuscript is written in a somewhat biased manner. Based on the authors' response, they are aware that trans-intoxication is not occurring for a reason independent of the presence or absence of the 2610 immunity protein. So to state the manuscript is testing whether 2610 functions to prevent cis-intoxication vs. trans-intoxication is misleading. This concept should be stated more clearly, ie they are only testing its role in preventing cis-intoxication. Therefore, Figure 2 probably should be a supplementary figure and not a main figure.

2. I disagree with the other reviewers that determining the mechanism of how the toxin gains access to the periplasm is important. The authors have tested the obvious things (e.g. dependence on the T4SS signal sequence and the T4SS). As a result, it is likely that a small amount of the toxin is mysteriously leaking into the periplasm by some unknown mechanism, which will be very difficult to impossible to decipher/demonstrate.

3. Figure 2 legend:

a. D should be C.

b. Not clear why there is a single p value in Fig. 2B when there are 3 different samples compared to WT(kanR).

Rev_Com_number: RC-2023-01856

New_manu_number: EMBOR-2023-58138V1

Corr_author: Farah

Title: The protective function of an immunity protein against the cis-toxic effects of a Xanthomonas Type IV Secretion System Effector

As requested by the Editor,

- We have reduced the number of characters in the title to fit within the limit of 100 characters with spaces. The new title is “Immunity protein protection against the cis-toxic effects of a *Xanthomonas* T4SS Effector”.
- We also reduced the Abstract to 175 words, as required.
- We have converted old Supplementary Figures 7, 10 and 11 into Expanded View Figures EV1, EV2 and EV3, respectively. The remaining Supplementary Figures, Tables and Data are now found in the single pdf file labeled “Appendix” containing Appendix Tables 1-5, Appendix Figures S1-8 and Appendix Data S1.
- Source Data for Figures 2, 3, 4, 5 and 6 have been provided as requested. Due to their large sizes, the micrographs used to produce Figures 3 and 5A have been deposited in the Biostudies archive: (<https://www.ebi.ac.uk/biostudies/S-BSST1204>). This is mentioned in the Data Availability statement of the revised manuscript.

Responses to the Reviewer's comments.

All reviewers recommended publication of the revised manuscript. Reviewers 1 and 2 are satisfied with the revised version. Reviewer 3 had a few outstanding comments that we have responded to below.

Reviewer 3: 1. Unfortunately, I still think the manuscript is written in a somewhat biased manner. Based on the authors' response, they are aware that trans-intoxication is not occurring for a reason independent of the presence or absence of the 2610 immunity protein. So to state the manuscript is testing whether 2610 functions to prevent cis-intoxication vs. trans-intoxication is misleading. This concept should be stated more clearly, ie they are only testing its role in preventing cis-intoxication. Therefore, Figure 2 probably should be a supplementary figure and not a main figure.

Our response: We are not being misleading on this point. During the course of the study we used an *X. citri* strain in which we deleted all nine different effectors (toxins) and eight of their cognate immunity proteins (anti-toxins) and these cells remained resistant to killing by wild-type *Xanthomonas citri* cells. This observation is inconsistent with the hypothesis that the immunity proteins protect against trans-intoxication (transfer of toxins from neighboring wild-type cells). On the other hand, the deletion of a single immunity protein causes *Xanthomonas citri* cells to undergo premature lysis. This was the first evidence that the immunity proteins are providing protection mainly against cis-intoxication, instead of trans-intoxication. We therefore maintained Figure 2 as a main Figure.

2. I disagree with the other reviewers that determining the mechanism of how the toxin gains access to the periplasm is important. The authors have tested the obvious things (e.g. dependence on the T4SS signal sequence and the T4SS). As a result, it is likely that a small amount of the toxin is mysteriously

leaking into the periplasm by some unknown mechanism, which will be very difficult to impossible to decipher/demonstrate.

We agree with this reviewer on this point. Determining the precise mechanism by which proteins without signal peptides gain access to the periplasm will have to be left to future studies.

3. Figure 2 legend:

a. D should be C.

b. Not clear why there is a single p value in Fig. 2B when there are 3 different samples compared to WT(kanR).

Our response: We have reanalyzed the data and corrected these issues in Figure 2 and its legend. The single p value for Figure 2B refers to the Anova test which shows that there are no significant differences between the means of all four samples. This is now clarified in the legend. We have also provided the unpaired t-test p values for the data in Figure 2C in the figure legend.

Dear Dr. Farah,

Thank you for the submission of your revised manuscript to our editorial offices. We are nearly done. Before we can proceed with formal acceptance, I have these editorial requests I ask you to address in a final revised manuscript:

- I would suggest a more active title:

A Xanthomonas immunity protein protects against the cis-toxic effects of a T4SS effector

- Please have your final manuscript carefully proofread by a native speaker.

- Please reduce the number of keywords to 5 and order the manuscript sections like this, using these names:

Title page - Abstract - Keywords - Introduction - Results - Discussion - Materials and Methods - Data availability section - Acknowledgements - Disclosure and Competing Interests Statement - References - Figure legends - Expanded View Figure legends

- Please make sure that the number "n" for how many independent experiments were performed, their nature (biological versus technical replicates), the bars and error bars (e.g. SEM, SD) and the test used to calculate p-values is indicated in the respective figure legends (main, EV and Appendix figures). Please also check that all the p-values are explained in the legend, and that these fit to those shown in the figure. Please provide statistical testing where applicable. Please avoid the phrase 'independent experiment', but clearly state if these were biological or technical replicates. Please also indicate (e.g. with n.s.) if testing was performed, but the differences are not significant. In case n=2, please show the data as separate datapoints without error bars and statistics. See also:

<http://www.embopress.org/page/journal/14693178/authorguide#statisticalanalysis>

If n<5, please show single datapoints for diagrams. Presently, some diagrams don't have statistics or only partial statistics (e.g. panels 2B, 2C, 4C and S5). Moreover:

- Although 'n' is provided, please describe the nature of 'n' in the legends of figures 2a, 6d and S5 (biological or technical).

- Please define the error bars in the legends of figures 2a-c and 6d.

- Please add scale bars of similar style and thickness to all the microscopic images (main, EV and Appendix figures), using clearly visible black or white bars (depending on the background). Please place these in the lower right corner of the images themselves. Please do not write on or near the bars in the image but define the size in the respective figure legend. Presently, there are scale bars outside the image and several with text nearby.

- Please make sure that all figure panels and Tables are called out separately and sequentially (main, EV and Appendix items). Presently, Supplementary File S1 and S1 Supplementary File are called out, but it is not sure what these callouts refer to. The callout for Fig. S2B needs to be corrected to Appendix Fig. S2B and the callout for Fig. S9 needs to be corrected to Appendix Figure S9. Please check.

- Please remove the title page from the Appendix file. It is sufficient to state there 'Appendix for ...', followed by the table of contents.

- The 'Appendix Data S1' shown in the Appendix file is a dataset. Please remove this from the Appendix and upload this as dataset file. Please upload the original excel file with a title and a legend on the first TAB. Please name this file Dataset EV1 and update its callouts.

- Please upload each movie file ZIPed together with a readme.txt file (with title and legend). Please name the uploaded files Movie EV1-EV7. Finally, please remove the movie legends from the manuscript text file.

In addition, I would need from you:

- a short, two-sentence summary of the manuscript (not more than 35 words).

- two to four short (!) bullet points highlighting the key findings of your study (two lines each).

- a schematic summary figure as separate file that provides a sketch of the major findings (not a data image) in jpeg or tiff format (with the exact width of 550 pixels and a height of not more than 400 pixels) that can be used as a visual synopsis on our website.

Best,

All editorial and formatting issues were resolved by the authors.

Dr. Chuck Farah
Universidade de São Paulo
Departamento de Bioquímica
Av. Prof. Lineu Prestes 748
São Paulo, SP 05508-900
Brazil

Dear Dr. Farah,

I am very pleased to accept your manuscript for publication in the next available issue of EMBO reports. Thank you for your contribution to our journal.

Yours sincerely,

Rev_Com_number: RC-2023-01856
New_manu_number: EMBOR-2023-58138V3
Corr_author: Farah
Title: A Xanthomonas immunity protein protects against the cis-toxic effects of its cognate T4SS effector